

# The Spatial Extent of Hydrological and Landscape Changes across the Mountains and Prairies of the Saskatchewan and Mackenzie Basins

Paul H. Whitfield[1,2,3], Philip D.A. Kraaijenbrink[4], Kevin R. Shook[1], John W. Pomeroy[1]

5  [1]Centre for Hydrology, University of Saskatchewan, Saskatoon, SK, S7N 1K2, Canada

[2]Department of Earth Sciences, Simon Fraser University, Burnaby, BC

[3]Environment and Climate Change Canada, Vancouver, BC

[4]Geosciences, Utrecht University, Utrecht, The Netherlands

*Correspondence to*: Paul Whitfield (paul.h.whitfield@gmail.com)



**Abstract** [486 words]

East of the Continental Divide, the cold interior of Western Canada has one of the world's most extreme and variable climates and is experiencing rapid environmental change. In the large Saskatchewan and Mackenzie River basins, the warming climate is changing the landscape, vegetation, cryosphere, and water. This study of a large number (395) of gauged basins in these large river basins provides the basis for a large-scale analysis of observed hydrological and landscape changes. In this region, the existing data sets are complex; observed streamflow records are available for differing series of years and streamflow measurements consist of both continuous and seasonal records. This diversity has been compensated for using novel analytical approaches: [1] a Streamflow Regime classification using dynamic time-warping that covers only the common period of the calendar year amongst all stations, and which is restricted to a time window of seasonal observations, [2] a classification of *seasonal* Streamflow Regime change using k-means clustering of the year divided into five-day bins. An assessment of landscape and hydrological storage change for each gauged basin was conducted using Landsat 5 TM imagery of Normalized Difference Vegetation, Water, and Snow Indices (NDVI, NDWI, and NDSI) for 1985 to 2010. Therefore, this analysis is for a different time period than the streamflow regime and trend patterns.

Twelve "Streamflow Regime Types" were identified using dynamic time-warping to overcome the problem of timing differences producing flow clusters due to latitude or elevation, rather than from the shape of the hydrograph resulting from differing processes. The success of this approach suggests that there is sufficient information in the time window to adequately resolve regions; Streamflow Regime Types exhibit a strong connection to location; the spatial distribution follows ecoregions and shows a strong distinction between mountains and plains. Clustering of seasonal trends resulted in six "Trend Patterns". The Trend Patterns also have a strong and distinct spatial organization. The Trend Patterns include one with decreasing streamflow, four with different seasonal increasing streamflows, and one without any apparent trend structure. Trends in the mean, minimum, and maximum of three satellite indices were determined; the spatial patterns of NDWI and NDSI were similar to each other, but NDVI patterns were generally dissimilar. Streamflow Regime Types, the Trend Patterns, and satellite indices showed spatial coherence. The overlap between hydrological and landscape change was not perfectly coherent, suggesting that landscape changes may have a different domain



from the existing hydrological regimes and from the observed trend patterns. Three particular areas of change are identified: [i] north of 60° where streamflow and greenness are increasing while wetness and snowcover are decreasing, [ii] in the western Boreal Plains where

streamflows and greenness are decreasing but wetness and snowcover are not changing, and [iii] across the Prairies where there are three patterns of increasing streamflow and an increase in the wetness index, the largest changes occur in the eastern portion of the Canadian Prairies, with only few increases in greenness and snow indices. The results demonstrate the spatial extent of these changes.

## 1. Introduction

The cold interior of Western Canada east of the Continental Divide has one of the world's most extreme and variable climates and is experiencing rapid environmental change (DeBeer et al 2016). Changing climate is affecting the landscape, the vegetation, and the water. The southern portion of this region sustains 80% of Canada's agricultural production, a large portion of its forest wood, pulp and paper production, and also includes several globally-important natural

resources (e.g. uranium, potash, coal, petroleum). Understanding observed changes and possible future changes is clearly in the national interest. Climate variation and change has been demonstrated to have important effects on the rivers of Canada (Whitfield and Cannon 2000; Zhang et al 2001; Whitfield et al 2002; Janowicz 2008; Déry et al 2009; Tan and Gan 2015) including Western Canada's cold interior (Luckman 1990, Burn 1994; Luckman and

Kavanaugh 2000; Ireson et al 2015; Dumanski et al 2015; Ehsanzadeh et al 2016). Botter et al (2013) demonstrate that the sensitivity may differ by period of the year. Trends in water storage based on Gravity Recovery and Climate Experiment (GRACE) Satellites, identified precipitation increases in northern Canada, progression from a dry to a wet period in the eastern Prairies/Great Plains, and an area of surface water drying in the eastern Boreal (Rodell et al





2018). In this study area, where the Mackenzie and Saskatchewan River Basins are the focus, cold region climatic, hydrological, ecological, hydrological, and cryospheric processes are highly susceptible to the effects of warming. This study provides a statistical assessment of patterns and recent changes in hydrological regime, water storage, snow and landscape processes and their spatial pattern at the gauged basin scale and across these basins.

Hydrological processes differ widely in this domain, which spans 11 of Canada's 15 ecozones, and includes many smaller basins where streamflow is only temporary (Buttle et al 2012). The hydrographs of all rivers in this domain reflect the contribution from snowmelt which differs in space and in time. There are important regional differences in how snowmelt contributes to the seasonal pattern of hydrographs. Other flow contributions, from glaciers and rainfall, vary
spatially within the domain with glacier contributions focussed on the highest mountain headwaters and rainfall contributions increasing as elevation and latitude decline.

Streamflow data in this domain is taken from stations that are operated either year-round or seasonally (MacCulloch and Whitfield 2012); seasonal stations generally providing records from April through October, because there is either no streamflow in the winter, or the channels
become completely frozen. Most previous studies have focused on only the continuous streamflow stations (e.g. Whitfield and Cannon 2000) and novel aspects of this study are that a method incorporating and linking seasonal stations and records from temporary rivers is demonstrated, and trend assessment is conducted on a common annual time window.

Landscape changes may cause or result from hydrological change. Satellite imagery and
derived spectral indices were used to assess the impacts of environmental changes in the selected basins on their hydrological response. Trend analyses determined the existence of changes in vegetation, water bodies and snow cover for large regions (e.g. Hall et al., 1995; Su, 2000; Hansen et al., 2013; Pekel et al., 2016). Normalized Difference Indices of vegetation, water, and snow (NDVI, NDWI and NDSI) were constructed using optical imagery from the
Thematic Mapper (TM) sensor (USGS, 1984) on board the Landsat 5 satellite. Although the temporal coverage of the indices differs from the hydrometric data, trends in vegetation, water, and snow across a large number of basins provides another perspective of change over the study domain.



The objective of the present study is to examine hydrological structure and change in seasonal
pattern by integrating different forms of data, which previously had only been treated
separately and individually, to diagnose hydrological structure and change in western Canada's
cold interior. By linking continuous and partial year data from a large number of hydrometric
stations using only warm season data three important questions are addressed with respect to
the study domain: [1] how are the hydrological types and processes distributed?, [2] how are
climate related trends and mechanisms distributed?, and [3] are some hydrological types, and
processes more susceptible to change over time? By examining trends in normalized difference
indices, NDVI, NDWI, and NDSI for basins over a common time period, [4] landscape change
that may be driving or following hydrological change can be examined.


## 2.0 Methods

### 2.1 Data

The hydrometric (streamflow) stations selected for this study are active, (i.e. are currently
monitored), 'natural' (i.e. their flows are not managed), and either continuous or seasonal with
more than 30 years of data. No attempts were made to use a consistent period of years - rather
all analyses used the entire periods of record. Three additional stations were included that did
not meet the 30 years of data criteria.  These stations are Changing Cold Regions Network
(CCRN) Water, Ecosystem, Cryosphere, and Climate Observatories (WECC): Marmot Creek,
Smith Creek, and Scotty Creek (Table 1), and including these stations provides a link to CCRN
process-based studies.  Streamflow data from a total of 395 stations (gauged basins) was
available; approximately one-half (192 or 49%) of which were operated on a seasonal basis.

Clustering and statistical analysis were restricted to the window of the year when both seasonal
and continuous data sets were available.  The clustering of annual streamflow time series was
done using dynamic time warping, DTW, (Berndt and Clifford 1994; Wang and Gasser 1997)
which measures similarity between time series that may vary in magnitude and timing by
aligning the two curves in time, essentially matching the shape of inflections to create clusters
(Whitfield et al, 2020).  Gauged basins were also grouped based upon seasonal patterns of





significant trends using k-means, which partitions observations into clusters with the nearest mean and is well suited to clustering of features such as patterns of significant differences
(Likas et al 2003; Steinley 2006; Agarwal et al 2016).

All analyses were performed with R (R Development Core Team, 2014); using packages `kendall` (McLeod, 2014), `CSHShydRology` (Anderson et al 2018), and `dtwclust` (Sarda-Espinosa 2017, 2018). Maps were plotted using `maps` and related packages. A threshold of 0.05 is used in tests of significance, and %5 is also used as an indicators than the
number of trends exceeds that expected by chance alone.

Table 1 near here

As the intention was to include data from as many stations as possible, the entire period of record from each of the 395 gauged basins were used and only the window where data is
available from every station during the year, from 19 April to 31 October was analyzed. Because the periods of record, rather than a common period, are used it is not possible to compare the magnitudes of trends among the stations. Instead, the analyses are restricted to determining the existence of significant trends in individual five-day periods in (five-day) periods 23 to 61 (of 73), as shown in Figure 1 and 2. The main panel of the Figure 1 uses
colour to show the magnitude of the flow for each day of each year. This streamflow record is described in detail by Whitfield and Pomeroy (2016; 2017). The upper panel shows the minimum, median, and maximum for each five-day period and blue (red) arrows indicate periods where there is a significant increasing (decreasing) trend in streamflow across the period of record. The directions of significant trends (i.e. positive or negative) were determined
and were used for clustering of change types. The panel on the right shows the time series of annual minimum, median, and maximum discharges. If there was a significant trend (Mann-Kendall $\tau$, $p \leq 0.05$) the series is coloured (red for decreasing, blue for increasing, black indicates no trend). Many of the stations in this study, such as 05BA001 Bow River at Lake Louise, Alberta, have streamflow records only during the warm season (Figure 2). It is evident in
Figure 2 that since 1968 the streamflow data has only been collected in the summer, but previously the data was collected through the entire year. In this case, trends over time in annual minima, median, and maxima are based only upon available data and this shows in the





time series with step changes. Trends of these types should be based only on years with complete record and are not addressed further. In the supplementary information, Figures S1-
S12 show up to four example hydrometric stations for each Streamflow Regime cluster as described below. Many of these plots show that the operation of stations have alternated between being seasonal and continuous (e.g. Figure 2). These also demonstrate the variability of period of record between stations.

---

Figure 1 near here

Figure 2 near here

---

**2.1 Streamflow Regime Types**

Statistical methods, such as k-means (Likas et al 2003; Steinley 2006) or self-organized maps (Kohonen and Somervuo 1998; Hewitson and Crane 2002; Kalteh et al 2008; Céréghino and Park 2009; van Hulle 2012), are unable to group hydrographs when they are not aligned in time
(Halverson and Fleming 2015). Across this study domain this is a difficulty as timing of snow accumulation and melt are strongly affected by both latitude and elevation.

To avoid the effects of gauged basin area and location, the five-day streamflow records were converted to Z scores, by subtracting the mean value and dividing by the standard deviation, of the series five-day. Early snowmelt at low latitudes and elevations resulted in some stations
having flow events preceding the sampled date range (Figure 3). Only the data in the periods between the two vertical dashed lines in Figure 3 were used in the clustering (and trend analysis) reported here.

---

Figure 3 near here

---

The clustering of flows into Streamflow Regime types five-day used dynamic time warping
(DTW) (Wang and Gasser 1997; Keogh and Ratanamahatana 2005) which measures similarity between time series that may vary in magnitude and timing. DTW aligns the two curves in time (Figure 4), essentially matching the shape of inflections. The R package `dtwclust` (Sarda-Espinosa 2017) implements dynamic time warping to cluster curves based upon their having similar shapes and inflections. The timing of inflections does not affect the clustering,
hence the effects of latitude and elevation that often result in misclassification of hydrographs because of timing differences are avoided, which is important in a spatial domain of the size



being considered here. A twelve cluster solution was chosen; this number of clusters balances regional separation of similar Hydrograph Types while avoiding producing many types with single stations which represent unique hydrological situations. Streamflow Regime Type in the text which follows refers to these twelve clusters.

---

Figure 4 near here

---

## 2.2 Trend Patterns

Trends in the five day DTW periods were determined for the period of record for all records ending in the same year, using Mann-Kendall tests. Since these were comparing periods separated by 360 days, autocorrelation was not expected and therefore pre-whitening was not applied. The trend test results are indicated in the upper panel of Figures 1 and 2 and S1-S12 where significant increases and decreases are indicated by blue and red arrows respectively. The significant increasing, null, and decreasing trends were assigned scores 1, 0, and -1 respectively. The annual trend scores were clustered was using k-means. The number of clusters chosen was based upon the elbow method (Ketchen & Shook 1996; Kodinariya & Makwana 2013); using more than six clusters did not improve the modelling (not shown). These six clusters are referred to as Trend Patterns in the following text.

---

Figures S1-S12

---

## 2.3 Landscape and Hydrological Storage Trends

### *Extraction of spectral indices*

Satellite imagery and derived spectral indices are valuable for assessing effects of environmental changes the hydrological responses of the gauged basins as these methods allow determination of changes in vegetation, water bodies and snowcover for large areas (e.g. Hall et al 1995; Hansen et al 2013; Pekel et al 2016; Su 2000). Time series of spectral remote sensing indices were constructed using optical imagery from the Thematic Mapper (TM) sensor (United States Geological Survey (USGS), 1984) on the Landsat 5 satellite for each gauged basin. The satellite had a 16-day return period for any location on the Earth's surface and acquired imagery with a spatial resolution of 30 m between 1984 and 2011. The sensor was selected for its spectral capabilities, which allowed evaluation of surface changes, and its long



operation which best suits the length of the hydrological record in the gauged basins. It was chosen to avoid combining data from different satellites or sensors, so as to maintain consistency in spectral response over the study period. Later satellites use different spectral bands.

Given the large size of the study region and the length of study period, analysis of time series of Landsat 5 TM data is very resource intensive, both in terms of storage and computational power, and is beyond desktop computing. Google Earth Engine (GEE) allows for cloud-based planetary scale analysis while it serves as a database for petabytes of open access satellite imagery such as the Landsat archive (Google Earth Engine Team 2017; Gorelick et al 2017),

and is particularly capable for this study.

Using GEE, Landsat 5 TM image composites ($n$=579) were produced that cover the extent of all the gauged basins at a 16-day interval for the period 1985–2010. This was performed by mosaicking the available image scenes for each consecutive 16-day period. Theoretically this would allow for spatially-complete mosaics given the 16-day revisit time of the satellite. To

determine accurate spectral indices at the basin-scale, pixels containing clouds or cloud shadows in each image scene were masked prior to mosaicking using the GEE-integrated Fmask algorithm (Zhu et al 2015), which introduced intermittent data gaps to the set of mosaics. Additional data gaps are caused by occasional unavailability of satellite image scenes in the Landsat 5 TM catalogue, typically due to image quality or georeferencing issues (USGS

1984). To reduce the computation cost and data volume, the final mosaics were generated at an aggregated spatial resolution of 300 m. The total number of satellite image scenes used was 83381.

For each basin, three time series of spectral index averages were derived from the 16-day mosaics of Landsat 5 TM data. Each of these three normalized difference indices compares

two wavelength ranges observed by the satellite detectors using the form I = (W1-W2)/(W1+W2) and each index ranges from -1 to 1. The three common indices used were the Normalized Difference Vegetation Index (NDVI), Normalized Difference Water Index (NDWI) and Normalized Difference Snow Index (NDSI) (Lillesand et al 2014), and are calculated by:

$$NDVI = (R_{NIR} - R_{Red})/(R_{NIR} + R_{Red}),$$ (1)




$$NDWI = (R_{NIR} - R_{SWIR})/(R_{NIR} + R_{SWIR}), \text{and} \tag{2}$$

$$NDSI = (R_{Green} - R_{SWIR})/(R_{Green} + R_{SWIR}), \tag{3}$$

where $R$ is the dimensionless top-of-atmosphere reflectance, Green is TM band 2 (green light 0.52 – 0.60 μm), Red is TM band 3 (red light 0.63-0.69 μm), NIR is TM band 4 (near infrared 0.76 – 0.9 μm), and SWIR is TM band 5 (shortwave infrared 1.55 – 1.75 μm). Because of the

presence of the masked cloud pixels and data gaps, the basin means are often only calculated from a fraction of the complete pixel set of the basin, this fraction was determined for every observed time step.

For each sixteen day period, the mean NDVI, NDWI, and NDSI and the fractional coverage were determined for each of 375 gauged basins for which a shapefile of basin boundaries was

available (20 basin boundaries were not available from Water Survey of Canada). An example of the dataset is shown in Figure S13. From these time series of indices a time series of annual maximum, mean, and minimum were determined for each of the normalized different indices and the fractional coverage.

Forkel et al (2013) demonstrated the annual variability of in NDVI time series and the effects

of using analysis methodologies. Verbesselt et al (2010, 2012) and de Jong et al (2018) used breaks for additive season and trends (bfast) to detect change, particularly phenological change, in satellite imagery; bfast iteratively estimates the time and number of abrupt changes within time series derived from satellite images. While this methodology has considerable appeal and has been used widely and successfully to assess change in target areas, it was impractical to

apply bfast here as it is difficult to summarize multiple changes in seasonality, trends, and breakpoints for three indices across 375 basins. A simpler approach, testing for simple trends in the mean, maximum, and minimum indices, using Mann-Kendall (McLeod 2015) avoids the rich complexity possible with bfast but illustrates that land changes, and hydrological storage changes are accompanying streamflow regime changes. Minimum NDSI values excluded zero

values.

Figure S13



## 3.0 Results

### 3.1 Streamflow Regime Types

The Streamflow Regime Types from the twelve-cluster solution are shown in Figure 5. Each
of the twelve plots contains a line for each gauged basin in that Type and the dashed line the
median of all members; the colour of the lines is based upon basin ID. The differences in the
shapes of the hydrographs between types is evident and demonstrates how the shapes of the
members within a Streamflow Regime Type are similar. The outlying cases (e.g. Streamflow
Regime Types 6, 7, and 8) are also evident.

---

Figure 5 near here

---

Example plots for each of these twelve Streamflow Regime Types are shown in Figures S1 to
S12 to illustrate the similarity and differences between the hydrographs of within the
Streamflow Regime Types. Streamflow Regime Type 1 basins are generally Rocky Mountain
basins that have strong snowmelt signals (Figure S1). Streamflow Regime Type 2 basins are
reflective of Prairie streams with spring snowmelt and long periods with low or zero flow
(Figure S2). Streamflow Regime Type 3 basins are in the Athabasca River Basin (Figure S3).
Streamflow Regime Type 4 (Figure S4) have both strong snowmelt and late summer
streamflow. Streamflow Regime Type 5 basins are predominately Boreal Plains (Figure S5).
Basins from Streamflow Regime Types 6-8 and 10 are unique (or nearly so) but are similar to
adjacent types (Figures S6, S7, S8, and S10). An interesting feature of these four types is that
peak events occur at different times in different years. Streamflow Regime Type 9 basins peak
later in the summer, and are generally smoother that other basins (Figure S9). Streamflow
Regime Type 11 basins have an early snowmelt peak and higher flows extending through into
the fall (Figure S11). Streamflow Regime Type 12 basins have an early snowmelt peak and
persistent high flows during summer and extends into the fall (Figure S12).

---

Figures S1-S12

---

The spatial locations of the twelve Streamflow Regime Types are mapped over ecozones in
Figure 6; there is a clear spatial organization and not a random pattern. This association is also



evident in Table 2. Two large-scale features are evident; similar types tend to be from the same
spatial areas, and some similar types follow along major rivers. Streamflow Regime Types 3
and 11 follow along rivers and Types 2 and 5 overlap. Streamflow Regime Types 1 (104
members) and 5 (148) occur in the greatest numbers of ecozones (8 and 6 respectively; Table
2).

Figure 6 near here.

Streamflow Regime Type 1 basins occurs in the Cordillera (Montane n=31 basins, Boreal
n=11, and Taiga n=1) and also the Boreal Plains (n=31), Taiga Plains (n=5), Boreal Shield
(n=13) and Prairies (n=11); the Type 1 hydrograph shows the melt period being relatively brief
with a long, slow recession (Figure 5). Streamflow Regime Type 5 basins are also common in
the Prairie ecozone (n=49) in the Boreal Plains (n=81) as well as along the Mackenzie River to
below Great Slave Lake; this hydrograph shows an earlier and briefer peak than Type 1 with a
rapid recession (Figure 5). Streamflow Regime Type 4 basins (n=10) are predominantly found
in the Montane Cordillera in the west (n=5) and in the Boreal Shield and Boreal Plains in the
east (n=2 each); the hydrograph shows a prolonged high flows during the melt period, with a
shorter recession with relatively large flows (Figure 5). Streamflow Regime Type 3 basins
(n=22) appear in the Boreal Plains (n=17), Boreal Shield (n=2) and Prairies (n=3) and
demonstrates the persistence of a mountain runoff signal along the Athabasca River as this
hydrograph contains the late melt signal from glaciers (Figure 5). Streamflow Regime Type 2
basins (n=85) are associated with the Prairie ecozone (n=57) and Boreal Plains (n=28), but this
patterns also occurs in the Southern Arctic (n=1) and Taiga Plain (n=3); this pattern has the
earliest snowmelt and the records often start with snowmelt already in progress (Figure 5).
Streamflow Regime Type 11 basins (n=12) are near the mouth of the Mackenzie in the Taiga
Plains and in the Hudson Plains along the Nelson River in Manitoba. Streamflow Regime Type
12 basins (n=4) are in the Boreal Cordillera. Streamflow Regime Types 6-8 basins (1 each)
and 9 (n=2) are located at the edges of ecozones. While this description is stated as explicit,
there is overlapping of types in space (particularly Streamflow Regime Types 2 and 5), and
cases where individual basins of a Type occur quite separately from each other (Types 9 and
12) as is evident in Figure 6.

Table 2 near here.





A caution is warranted here. The shapes of hydrographs are controlled by climate, hydrological processes and landscape predominantly in the area where the flow is generated, and while association with an ecozone is useful, the ecozone is a generalization that does not always capture features in source areas that strongly affect hydrographs. In many rivers here, the source of flow is in the mountains and those patterns are transmitted along the downstream

path such as in the Mackenzie, Athabasca, and Nelson Rivers. Differences between Streamflow Regime Type 1 and 4 basins likely reflect low late summer flows from parts of the Rocky Mountains. In areas where there is great overlap between different Streamflow Regime Types, both similarities and differences exist between the basins.

**3.2 Trend Patterns**

Streamflow Regime trend patterns are based upon the statistical trends in five day flows discussed above. Figure 7 illustrates the trend results for one station, 05DA007 Mistaya River near Saskatchewan Crossing. In the case of Figure 7 there are three periods (39, 40, and 46) having significant decreases; the other periods have no trends.

Figure 7 near here

Figure 8 plots all significant trends for the 395 basins, with significant increase and decreases shown in blue and red, no trend in gray, and no available data in white. These data were ordered by the six Trend Patterns determined using only the data for the five-day periods from 23 to 61; Figure S14 shows the data order by stationID. The periods 1-22 and 62-73 were not

included in the clustering, but were plotted as they may also be of interest.

Figure 8 near here.

Figure S14

Trend Patterns in Figure 8 are presented in the order that clusters were formed, showing the distinctly different Trend Patterns 1, 2, 3, 4, and 6, while Trend Pattern 5, the largest group

with more than 250 basins (64%) of the total, does not have a consistent organized change. Trend Pattern 1 shows positive trends in most of the sampled window. Trend Pattern 2 has positive trends but only after period 30. Trend Pattern 3 has predominantly significant negative





trends, with many more in periods 40-60 than in periods 23-39. Trend Pattern 4 has significant increases centred about period 35. Trend Pattern 6 shows significant increases in the first
periods and last periods of the window but not in the summer; this group of stations all have winter data and show positive trends throughout the winter period.

Associating a trend pattern with a Streamflow Regime type is complicated as there are different numbers of basins in the Streamflow Regime Types and Trend Patterns (Table 3). Trend Pattern 5, which lacks a pattern, is very prominent in most of the Streamflow Regime types
having many members. Again, the caution regarding rivers sourced in the mountains and continuing downstream propagating the upstream signal also applies to their trend pattern as well.

Table 3 near here.

The six Trend Patterns of the 395 hydrometric stations are mapped to ecozones in Figure 9.
Supplementary Figures S15-S20 provide more detail on each of the Trend Patterns and the station locations. Trend Pattern 1 (n=19, S15) basins are located in the Prairies at the eastern and western margins. Trend Pattern 2 (n=22, S16) basins are located across the Prairies and in the eastern portion of the Boreal Plain. Trend Pattern 3 (n=32, S17) basins are located in the western portions of the Boreal Plain and in the eastern edge of the Montane Cordillera. Trend
Pattern 4 (n=50, S18) basins are located across the Prairies largely along the edge adjacent to the Boreal Plains and in a few northern locations in the Boreal Plain. Trend Pattern 5 (n=254, S19) does not show an organized change pattern in the window where all station have data are distributed across all ecozones. Trend Pattern 6 (n=18, S20) basins are located entirely in northern areas in the Taiga Plains, Taiga Shield, and Taiga and Boreal Cordillera. Overall,
28% of the basins show one of the four increasing trend patterns and 8% the single decreasing pattern.

At the ecosystem scale, 51% of basins in the Prairies exhibit a definite Trend Pattern with 45% showing one of the increasing patterns (Trend Patterns 1, 2, 4, and 6) and 6% the decreasing Trend Pattern 3 (Table 4). In the Taiga, increasing Trend Patterns predominate with 46% of
stations in the Taiga Plains showing increasing patterns and none with the decreasing Trend Pattern, 29% of the Taiga Shield having an increasing Trend Pattern and 14% a decreasing Trend Pattern, and all of the Taiga Cordillera (3 basins) having increasing Trend Patterns



(Table 4).  The Boreal Shield and Plains have increasing Trends Patterns in 16% of stations, and decreasing Trend Patterns in 13%.  The Boreal and Montane Cordillera have increasing

Trend Patterns in 11% of basins and only the Montane Cordillera had the decreasing Trend Pattern (5%) (Table 4).  Trend Pattern 6 only occurs in the northern portion of the study area. None of the stations on the Hudson Plains showed any change pattern.  This indicates that there is a spatial basis for the Streamflow Regime change that is influenced by location and ecozone rather than by Streamflow Regime Type.

Figure 9 near here.

    Table 4 near here.

    Figure S15-20

### 3.3 Landscape and Hydrological Storage Trends

The trends in the mean values of the three normalized difference indices are presented in Figure 10 and Tables 5 through 7.  The values for the trends in the maximum and minimum indices are plotted in the supplementary Figures S21-S23, and are also summarized in Tables 5 through 7.  The tables show the fraction of stations in a grouping that are significant at $p \leq 0.05$.  In all figures, trends with significance $\leq 0.05$ are shown as red (decreasing) or blue (increasing)

triangles, trends whose significance is less than 0.1 are shown as red or blue dots, and those with no trend in black.  There is a stronger association of the trends in the three indices with spatial location and with ecozones than with Streamflow Regime type or trend pattern. Frequently, the landscape and hydrological storage trends occur in a spatial domain that spans two or more ecozones (Figure 10, Figure S21 to S23).

Figure 10 near here.

    Tables 5, 6 and 7 near here.

    Figures S21-S23

    *NDVI*



The fraction of stations having statistically significant trends in NDVI (landscape trends) are greater than the 5% expected by chance alone across most Streamflow Regime Types (Table 5). Table 5 shows the combination of increasing and decreasing trends and each separately. For example, the fraction of all significant trends for maximum NDVI exceeds 5% in Streamflow Regime Types 1, 2, 3, 5, 9, and 10. The fraction of significant decreasing trends

for maximum NDVI is greater than 5% for Streamflow Regime Types 1, 3, 5, 9 and 10. The fraction of increasing trends in maximum NDVI exceeds 5% in Streamflow Regime Types 1, 2, and 5. All significant trends in mean NDVI are increasing and occur in Streamflow Regime Types 1, 9, 11, and 12. Significant trends in minimum NDVI are predominately decreasing in Streamflow Regime Types 2, and 5 and increasing in Streamflow Regime Types 1, 11 and 12

and both increasing and decreasing trends in Streamflow Regime Type 4.

There are also more significant trends in NDVI than would be expected by chance alone across all Trend Patterns (Table 6). The fraction of basins with significant trends in maximum NDVI exceeds 5% in all Trend Patterns, except Trend Pattern 3 which had no basins with significant trends. All mean NDVI trends are increasing and occur only in Trend Patterns 1, 4, and 6.

Minimum NDVI trends are predominately decreasing in all six Patterns and increasing in Trend Patterns 4 and 6.

The association of trends in mean NDVI between 1985 and 2011 with ecozone are each shown in Figure 10a (Figure S21) and Table 7. There is a stronger association of NDVI trends with ecozones than with either Streamflow Regime Types or Trend Patterns. Increasing trends in

mean NDVI occur in the Taiga Plains, Taiga Shield, Boreal Shield, Boreal Cordillera and Taiga Cordillera ecozones, decreasing trends in mean NDVI are found more often than expected by chance alone in the Montane Cordillera (Figure 10a, Table 7). The spatial patterns of trends in mean NDVI are similar to those of the maximum and minimum NDWI; however, there are more basins with significant trends in maximum and minimum NDVI than for mean NDVI.

Basins with significant increasing trends in maximum NDVI are found in the western portion of the Prairies and in the Boreal Shield (Figure S21a, Table 7); decreasing trends in maximum NDVI are found in the southern portion of the Boreal Plains the eastern Prairies, and the southern Boreal Plains and Boreal Shield (Figure S21a). The Taiga Plains has basins with both significant increasing and decreasing trends in minimum NDVI, decreasing trends were found

at greater than expected by change alone in Taiga Shield, Boreal Plains, and Prairies (eastern);



increasing trends were found in the Boreal Shield, Montane Cordillera, Boreal Cordillera, and Taiga Cordillera (Figure S21c). Basins with significant trends are not randomly distributed through any ecozone, as spatial clustering is evident in each of the three NDVI trends in Figure S21.


*NDWI*

There are more significant trends in NDWI (hydrological storage trends) than would be expected by chance alone in most Streamflow Regime Types (Table 5) and this is more prominent in the mean and minimum NDWI than in maximum NDWI. The fraction of

significant trends in maximum NDWI exceed 0.05 in Types 2, 5, 9, 11, and 12 with Streamflow Regime Types 9, 11, and 12 showing increasing trends much greater than the threshold (>0.20), while Types 2, 3, and 5 show decreasing trends but only near the threshold value. Significant trends in mean NDWI include both decreasing in Streamflow Regime Types 1, 3, 4, 9, 11 and 12 and increasing in Streamflow Regime Types 2, 3, and 5. Significant trends in

minimum NDWI are decreasing in Streamflow Regime Types 9 and 11 and increasing in Streamflow Regime Types 2 and 12 with both increasing and decreasing trends in Streamflow Regime Types 1, 3, 4, and 5. The largest fraction of significant trends ($> 0.33$) are decreasing trends in mean NDWI in Streamflow Regime Types 9, 11, and 12.

There are more significant trends in NDWI than would be expected by chance alone for all

Trend Patterns (Table 6). Significant decreasing trends in maximum NDWI exceed the threshold fraction in Trend Patterns 1 and 6, and increasing trends in Trend Patterns 3 and 4 (Table 6). Decreasing trends in mean NDWI exceed the threshold in Trend Patterns 1 and 6, as do increasing trends in Trend Patterns 1 to 5. The fraction of basins with decreasing trends in minimum NDWI exceeded the threshold in Trend Pattern 5 and for increasing trends in

Trend Patterns 1, 4, 5, and 6. Only Trend Pattern 5, i.e. without a trend pattern, was found to have both increasing and decreasing trends in minimum NDWI.

The association of trends in mean NDWI with ecozones are shown in Figure 10b and Table 7 (Figure S22 shows results for maximum, mean, and minimum NDWI). Similar to NDVI, there is a stronger spatial association of NDWI trends with ecozones than with either Streamflow

Regime Types or Trend Patterns. Decreasing trends in mean NDWI occur in the northern





ecozones (Taiga Plains, Taiga Shield, Boreal Cordillera and Taiga Cordillera; Figure 10b); increasing trends in mean NDWI occur only in the Prairies and Boreal Plains. There are more significant trends in mean NDWI than either maximum or minimum NDWI, but the spatial patterns for mean NDWI trends are similar to those for maximum and minimum NDWI. Basins

with significant increasing trends in maximum NDWI are found only in the western portion of the Prairies and the Boreal Shield (Figure S22a, Table 7); decreasing maximum NDWI trends are found in the eastern portion of the Boreal Plains, and in the Cordillera (Figure S22a). Both increasing and decreasing trends in minimum NDWI are found in the Taiga Plains, Boreal Shield, Boreal Plains, and Montane Cordillera (Figure S22c). Only decreasing trends were

found in the Taiga Shield and Hudson Plains, and increasing trends in minimum NDWI only occurred in the Prairies and Boreal Cordillera (Figure 22c). Basins with significant trends are not randomly distributed through any ecozone, as some spatial clustering is evident in each of the three NDWI trends in Figure S22.

*NDSI*

There are also more significant trends in NDSI (hydrological storage trends) than would be expected by chance alone across Streamflow Regime Types (Table 5) and this is most prominent in the minimum NDSI. Significant decreasing trends in maximum NDSI exceed the 0.05 threshold in types 9, 11, and 12; increasing trends exceed the threshold in Types 2, and 5.

Only decreasing trends in mean NDSI were detected in Streamflow Regime types 1, 4, 9, 11, and 12. Both decreasing and increasing trends in minimum NDSI occurred in Streamflow Regime types 1, 3, 4, and 11; in Streamflow Regime Type 9 only decreasing trends in minimum NDSI were found, and only increasing trends in Types 2, 10, and 12. The fraction of stations showing increasing trends in minimum NDSI is much greater than decreasing trends.

There are more significant trends in NDSI than would be expected by chance alone across all Streamflow Regime trend patterns (Table 6) with increasing and decreasing having similar frequencies. The fraction of significant decreasing trends in maximum NDSI exceed the threshold in Streamflow Regime trend patterns 1 and 6 (as was true for NDWI), and increasing trends in Trend Patterns 3, 4, and 5 (Table 6). No increasing trends were found in mean NDSI.

Decreasing trends in maximum and mean NDSI exceed the threshold in Trend Patterns 1 and


6, and minimum NDSI in Trend Pattern 5. Increasing trends minimum NDSI occurred in Patterns 1, 5, and 6. Only Trend Pattern 5, i.e. the pattern without a Streamflow Regime trend was found to have both increasing and decreasing trends in minimum NDSI.

The association of trends in NDSI with ecozones are each shown in Figure 10c, Figure S23, and Table 7. As with NDVI and NDWI, there is a stronger association of NDWI trends with ecozones than with Streamflow Regime Types or Trend Patterns. Decreasing trends in mean NDSI occur in the Taiga Plains, Taiga Shield, Boreal Shield and Boreal Cordillera (Figure 10c, Table 7); there were insignificant increasing trends in mean NDSI, but significant increases are evident in Figure 10c in the Prairies and Boreal Plains. There are many more significant trends
in minimum NDSI than in either maximum or mean NDSI (Table 7, Figure S23). The spatial patterns for mean NDSI are similar to those for maximum and minimum NDSI. Basins with significant decreasing maximum NDSI trends are found in the Taiga Plains, Taiga Shield, and Boreal Cordillera (Figure S23a, Table 7); decreasing maximum NDSI are found in the eastern portion of the Boreal Plains, and in the Prairies. Trends in minimum NDSI are more numerous
than for maximum or mean NDSI (Table 7) and include both increasing and decreasing trends in the Taiga Plains, Boreal Shield, and Boreal Plains (Figure S23c). Only decreasing minimum NDSI trends were found in the Montane Cordillera, and increasing trends in minimum NDSI only occurred in the Prairies, Boreal Cordillera, and Taiga Cordillera. Basins with significant trends are not randomly distributed through any ecozone, as spatial clustering is evident in each
of the three NDSI trends in Figure S23.

### 3.4 Trend Patterns and Landscape Change

Trend pattern 6 (Figure S20), with prominent winter increases in streamflow across the north (Taiga and subarctic) was originally described by Whitfield and Cannon (2000). This region
also has significant increases in mean NDVI (Figure 10a), and decreases in mean NDWI and NDSI (Figure 10b). At this scale, one possible interpretation of the observed trends would be that warming has altered the seasonal pattern of permafrost and increased winter flows and also the greenness of these basins which suggests higher evapotranspiration, (Figures 10a and S21) and reduced the amount of standing water (Figures 10b and S22), and the snowcovered period
(Figures 10c & S23).



Three other increasing trend patterns (1, 2, & 4) show different temporal patterns of change. These three patterns were observed on the Prairies and southern Boreal Plains. The other three increasing streamflow change patterns occur across the Prairies and the southern edge of the Boreal Plains. These three Trend Patterns have considerable spatial overlap (Figure 9).

Trend Pattern 1 (Figure S15) is common in the Manitoba portion of the Prairie and Boreal Plains ecozone, with greater streamflows throughout the period when data were available. This region also has no significant changes in mean NDVI (Figure 10a) nor in NDSI (Figure 10c), but there are increases in NDWI at the western edge of the region (Figure 10b). One possible interpretation is that landscape and hydrological storage changes have been subtle, streamflows

have increased without altering the greenness and hence evapotranspiration of these basins (Figures 10a and S21) or the snowcovered period (Figures 10c & S23) but have increased wetness in the western side of this region (Figures 10b and S22).

Trend Pattern 2 (Figure S16) is common in the Alberta and Saskatchewan portion of the Prairie, and the Saskatchewan and Manitoba portion of the southern Boreal Plains ecozone. This region

also has no significant trends in mean NDVI (Figure 10a) but does display significant positive trends in mean NDWI (Figure 10b) and NDSI (Figure 10c). A possible interpretation is that landscape changes have not altered the greenness of these basins (Figures 10a and S21) but that the satellite images have showed an increasing trend in wetness (Figures 10b and S22) which may be the result of increases in summer precipitation, and an increase in the

snowcovered period which may be due to increased winter precipitation (Figures 10c & S23).

Trend Pattern 4 (Figure S18) is common across the Prairie and southern Boreal Plains ecosystems. The region generally has no significant trends in mean NDVI (Figure 10a) and only a few increasing trends in NDSI (Figure 10c), which are restricted to Saskatchewan, but with significant increases in mean NDWI (Figure 10b) except in Manitoba. An interpretation

is that landscape alterations have not altered the greenness of these basins (Figures 10a and S21) but that the satellite images are showing an increase in wetness (Figures 10b and S22) which may be the result of increases in summer precipitation, and an increase in the snowcovered period which may be due to increased winter precipitation (Figures 10c & S23).

Trend Pattern 3 (Figure S17) is the only change pattern with decreasing streamflows and is

common across the Prairie and southern Boreal Plains ecosystems. Decreasing trends in



streamflow are more prevalent in the latter portion of the observed period. The region in which this pattern is observed is concentrated in the western part of the Boreal Plain and into the western portion of the Prairie ecosystem. This region has generally seen significant decreasing trends in mean NDVI (Figure 10a) and the overlap of the areas with Trend Pattern 3 basins

corresponds well with those basins demonstrating significantly decreasing mean NDVI and without trends in NDWI (Figure 10b), or in mean NDSI (Figure 10c). One interpretation could be that the greenness of these basins (Figures 10a and S21) has decreased because the basins are drier in summer but satellite images do not indicate decreases in wetness (Figures 10b and S22) or snowcover (Figure 10c and S23).


**Discussion**

Trends in streamflow due to climate or landscape changes are not expected to be spatially uniform (Carey et al 2010; Patterson et al 2012). Many studies that seek to explore change perform many analyses simply to see what falls out. The approach used here is targeted to

assess which aspects are changing and why, the basic element being the hydrological response that depends upon the key streamflow-generating processes in each basin. Analyses of change in this analysis are focused on seasonal patterns rather than on annual measures. Given the large datasets, where stations have records for differing years, with a large number of basins that only having warm-season data, determining the magnitude of trends in annual runoff, and

other annual attributes is too complex. In such a large spatial domain, it is essential that spatial linkages and patterns are considered for both Streamflow Regime types and change patterns. By focusing on warm-season data and the seasonal pattern in flows and in their trends, their complex spatial structure and trend patterns can be explored. Using as many stations as possible improves the resolution of the spatial extent of changes.


**4.1 Streamflow Regime types**

Bower et al (2004) argue that streamflow regimes are a useful way of considering seasonal hydrology. Unfortunately, the large numbers of standardized streamflows plotted in Figure 5 make it difficult to compare the Streamflow Regime Types; plotting the z-score centroids of


each (Figure 11) makes the comparisons clearer. Each of the Streamflow Regime Types centroids have a dominant peak, and a specific shape. The peaks may be narrow (Types 1 and 11) or broad (Types 3, 4, 9 and 12). Streamflow Regime Types 2 and 5 associated with the Prairies were not always sampled on the rising limb, because a) the peaks are due to snow melt, b) the Prairies melt early in the year, and c) the snow melt period often occurs before the

beginning of the annual analysis period.

The non-prairie Streamflow Regime Types have approximately linear recessions with two (Types 1, 3, 4, and 12) or more slopes (Type 11). The slopes of the recessions are listed in Table 8. Typically the first recession phase is steeper than the second phase where there is one. In Streamflow Regime Type 11 the third phase is steeper than the second, but not as steep as

the first. After a rapid recession, Streamflow Regime Type 2 becomes nearly horizontal, probably due to Prairie streams typically having no base flows. In Streamflow Regime Type 5 the recession has two linear phases, and also terminates in a horizontal section, which is again likely to be caused by the absence of base flows in the Prairie stations. These recessions appear to be limited to only five values, two that are steep (-0.22 and -0.16) and two that are much

flatter (-0.06 and -0.04) as well as the zero slope of Type 5.

Figure 11 near here

Table 8 near here.

Sample hydrographs of individual stations from each type, are shown in Supplementary Figures S1-12. These similarities among the stations are due to the predominance of snowmelt as the

source of streamflow in this cold region, and the differences are related to the combinations of landscape processes that mediate the melt (hypsometry, surface storage, groundwater, and glaciers). The classification produced by the algorithm is spatially reasonable in that stations from similar landscapes, hydrology, and climates are clustered together (Table 2). Stations which are nested along a single basin (e.g. Athabasca River, Type 3) are clustered together as

the signal derived from the mountains propagates downstream. Distinctions between clusters draining differing terrains, which overlap spatially, such as at the boundary between the Cordillera and Prairies, are to be expected as the landscape gradients or differences are large. Similar results were found in Ontario (Razavi & Coulibaly 2013).





Three clusters (Streamflow Regime Types 6, 7, & 8) each contained a single member and one

(Type 10) had only two members. These Streamflow Regime Types are very different from those containing large numbers of members and any clustering of hydrographs in this way balances common characteristics against uniqueness.

### 4.2 Trend Patterns

Like the Streamflow Regime Types, the trend patterns also show strong spatial organization; one pattern of decreasing trends, four types of increasing trends, and a large group without trends (Table 3). There is no clear link between the Streamflow Regime Type and the Trend Pattern (Table 3). Basins in mountainous areas generally lack consistent patterns of trends. The Trend Patterns have spatial distributions that appear unrelated to individual ecozones but

as would be expected from past descriptions of the predominant climate processes across the Prairies, drying in the west and wetting in the east (Borchert 1950; Rosenberg 1987; Luckman 1990, Whitfield et al., 2020), and extend beyond any one ecozone. For example, across the Prairie ecozone (Figure 9), there are three increasing Trend Patterns. Trend Pattern 1 basins are located at the eastern and western margins of the Prairies (Figure 9, Figure S15). Trend Pattern

2 basins are located across the Prairies but are concentrated to the east and into the eastern portion of the Boreal Plains (Figure 9, Figure S16). Trend Pattern 4 basins are scattered across the Prairies and occur along the southern margins of the Boreal Plains (Figure 9, Figure S18). The decreasing Trend Pattern 3 basins are prominent in the region just east of the Rocky Mountains across the Boreal Plains and just west of the Prairies (Figure S17).

While annual trends in climate and streamflow have been widely reported (e.g. Zhang et al 2001; Peterson et al., 2002; Déry et al., 2009a; St. Jacques and Sauchyn, 2009, Shook and Pomeroy, 2012), seasonal studies are less common, although at finer scales of analysis process shifts can be determined (Whitfield and Cannon 2000; Whitfield et al., 2002; Bennett et al 2015; Auerbach et al 2016). The focus of the present study is on runoff timing changes

using methods originally developed by Leith and Whitfield (1998) and subsequent improvements (Déry et al 2009b). Many of the changes reported here have been observed by others. Increased winter streamflow in northern Canada was reported by Whitfield and Cannon (2000) and others subsequently (St. Jacques and Sauchyn 2009; Bawden et al 2015).



Timing shifts, particularly in the onset of spring freshet were studied by Burn 1994,

Westmacott & Burn 1997) and lower summer recessions were studied by Leith and Whitfield 1998. The changes in basins on the Boreal Plains and Prairies are novel findings as studies in the past have not incorporated data from hydrometric stations with seasonal records. Using a classification of annual runoff patterns in streams across the Prairies and adjacent areas (Whitfield et al., 2020) determined that basins in the east portion of the Prairies becoming

wetter, and basins in the west becoming drier was expected (Borchert 1950), and consistent with the increased streamflows found at Smith Creek (Dumanski et al 2015). But, no changes in precipitation or runoff of seasonal and continuous streamflow records were detected by Ehsanzadeh et al (2016). Several studies of precipitation in summer across the Prairies reflect a similar pattern changes (Akinremi et al 1999; Asong et al 2016; DeBeer et al

2016). In the US Great Plains, Chatterjee et al (2018) identified an intensified drying and increased number and duration of low flow periods and higher flow events of shorter duration.

The spatial clustering of trends does not coincide with Streamflow Regime types, or with ecozones. The contiguous change regions are broad in space and span across ecozone

boundaries. This is indicative of changes that are taking place at scales different from those that generate runoff (Streamflow Regime type), but are related to broad scale changes that would be expected with changes in weather patterns across the southern portion of the study area and with climate warming in the northern portion. These patterns were partially described by Whitfield and Cannon (2000), and they have been reported on and explored by others since

(Burn and Hag Elnur 2002; Woo and Thorne 2003; Fleming 2007; Janowicz 2008; Bawden et al 2015; Tan et al 2017).

### 4.3 Landscape and Hydrological Storage changes

The primary interest in landscape and hydrological storage change in this study is to determine

if changes, as evidenced by trends in normalized difference indices (NDVI, NDWI, and NDSI) observed over a common time period indicate landscape or hydrological changes that may be driving or following streamflow regime change. Changes in land use and hydrology at the basin scale drive streamflow regime changes (Fohrer et al 2001; Woo et al 2008). Satellite imagery is increasingly being used for basin-scale analysis, but these generally focus on a small





number of cases (Coppin et al 2004; Bevington et al 2018; Soulard et al 2016; Militino et al
2018; Jorgenson et al 2018; Lee et al 2018).

The results presented here consider spatial patterns of hydrological, streamflow and landscape
changes that show several broad patterns, which warrant closer examination in the future. Four
specific regions have results that demonstrate significant changes, or lack thereof, in hydrology
and landscape indices. These four correspond to identified trends in global water storage based
on Gravity Recovery and Climate Experiment (GRACE) Satellites (Rodell et al 2018);
precipitation increases in northern Canada, a progression from a dry to a wet period in the
eastern Prairies/Great Plains, a region of surface water drying in the eastern Boreal, and no
changes along the Rocky Mountains. While the GRACE trend regions described by Rodell et
al (2018) are consistent with the results presented here, the mechanisms they suggest may not
be.

*North of 60°*

Warming has increased winter flows (Figure 8 and S20) (Whitfield et al 2004) and increased
the greenness of these basins (Figures 10a and S21) and reduced the amount of standing water
(Figures 10b and S22), and the snowcovered period (Figures 10c & S23). Climate change
might affect Arctic hydrology by decreased snowcover and snowfall in space and time,
decreased depths of soil freezing; increased snow-free season rainfall; northward movement of
the southern limit of permafrost; and, earlier snowmelt with more frequent ice lenses (Kane
1997) however it may also increase snowfall and snowcover (Krogh and Pomeroy, 2019). In
northern Canada, regional and local hydrology are controlled by permafrost thaw (Whitfield
and Cannon 2000; Cannon and Whitfield, 2002) by increasing surface and subsurface
connectivity, and by baseflow (Connon et al., 2014; Liljedahl et al., 2016; Carpino et al 2018;
Quinton et al 2018). Annual runoff in the Mackenzie River has increasing trend in due to
increasing annual flow trends in the Liard and Peace Rivers (Rood et al 2017). Rodell et al
(2018) attributed the increasing trends in northern Canada to freshwater accumulations (e.g.
Forman et al 2012).

Increasing trends in MODIS temperature and moisture patterns were more common that
decreases (Potter and Crabtree 2013). Changes in streamflow timing and peak have occurred
in snowmelt dominated streams of Boreal Alaska, with increasing winter flows, freshet flows,

and decreasing flows post-freshet similar to observed here (Bennett et al 2015). Climate driven changes in one taiga/tundra basin include decreasing snowfall, sublimation, soil moisture and evapotranspiration, deeper active layer, and an earlier loss of snow cover (Krogh & Pomeroy 2018). The results presented here indicate increased streamflow, particularly in winter, and basin wetness in a wide region north of 60°; however, the number

of basins available remains small.

*Boreal*

In the Boreal Plains, there has been more warming in winter and spring than in summer; none of the climate stations examined showed significant trends in annual precipitation over the interval 1950-2010, but some exhibited a significant decline in annual snowfall and snowfall

fraction due to a shortened cold season and earlier snowmelt (Price et al 2013; Ireson et al 2015). In the western Boreal Plains there has been a pervasive decrease in warm season streamflow (Figure 8 and S17) accompanied by a decrease in the greenness of these basins (Figures 10a and S21) because the basins are drier in summer although satellite images do not show a change in wetness (Figures 10b and S22) or snowcover (Figure 10c and S23). The

Boreal Plains is expected to be a region of strong ecological sensitivity under a changing climate (Ireson et al 2015). The southern margins of the Aspen Parkland and Boreal followed the climatic moisture suggesting that moisture supplies limit the southern extent of the forest (Hogg 1994). The driest areas of the Boreal forest are found at low elevations in west-central Manitoba, across Saskatchewan and Alberta and the southwest Northwest Territories; the

boundary follows the zero isoline of the moisture index (precipitation minus potential evapotranspiration) (Hogg 1994). These areas are drying with increased forest fire risk (Grosiman et al 2004). The Boreal Plains will become drier and have increased frequency of vegetation shifts and disturbances; forests are expected to contract in the north while the southern margin will remain stationary (Ireson et al 2015). Chronic moisture deficits are the

controlling factor of the boundary between forest and grassland in western Canada (Hogg 1997). In the Boreal Plains, evapotranspiration and changes in soil storage dominate water the balance (Devito et al 2005). Rodell et al (2018) suggest that the decreasing trend in "central Canada" [actually their region would be the western Boreal], were attributed to snow cover declines and recent drying (Bouchard et al 2013).


The subarctic climate of the Boreal region has large inter-annual variability and will be prone to future climate change (Woo et al 2008); models suggest that future winter flows will increase, spring melt will advance, but peak and summer flows will decline because ET will have increased (Devito et al 2005). The results presented here indicate widespread drying trends in the western portion of the Boreal Plains.

*Prairies*

Many studies of the Prairies published before 2010 showed drying trends, while more recent literature reports wetting areas. The climate of the prairies have become warmer and drier in the previous 50 years, and summer streamflows have decreased (Gan 1998). More recent literature demonstrates recent increases in precipitation; e.g. Gerken et al (2018) describe the 755 increases in precipitation and deceases in temperature in the Canadian Prairies during summer and demonstrate how this has increased to probability of convection and increased atmospheric moisture during summer. Garbrecht et al (2004) report increasing trends in precipitation, streamflow and evapotranspiration on the Great Plains; a 16% increase in precipitation led to a 64% increase in streamflow and occurred in fall, winter, and spring. The seasonality and 760 timing of streamflow in the north-central United States (Missouri, Souris-Red-Rainy, and Upper Mississippi Basins) have changed, i.e. northern portions have earlier snowmelt peaks and the probability of summer and fall peaks in streamflow has increased (Ryberg et al 2015).

Three streamflow Trend Patterns (1, 2, & 4) show increasing streamflows with different temporal patterns of change (Figure 8 and S15, S16 and S18) and overlap the Prairies and 765 southern Boreal Plains (Figure 9). Change Pattern 1 basins (Figure S15) are located in the eastern Prairies, and the adjacent Boreal Plains and are accompanied by increases in NDVI and no changes in NDWI or NDSI. Change Pattern 2 (Figure S16) is common in the Alberta and Saskatchewan portion of the Prairie, and the Saskatchewan and Manitoba portion of the southern Boreal Plains ecozone and is accompanied by increases in NDWI and NDSI but not 770 by changes in NDVI. Change Pattern 4 (Figure S18) is also common over the Prairie and southern Boreal Plains ecosystems. This region also has few significant changes in mean NDVI (Figure 10a) or NDSI (Figure 10c), but increases in NDWI at the western edge of the area (Figure 10b). This region also shows no significant changes in mean NDVI (Figure 10a) or NDSI (Figure 10c) but has significant increases in mean NDWI (Figure 10b). These changes 775 are complex and exhibit a series of changes in streamflow, greenness and snowcover from east





to west. Rodell et al (2018) suggest that the apparent wetting trend on the northern Great Plains arises partly because of the drought that took place before GRACE was deployed followed by years with above normal precipitation. Whitfield et al. (2020) demonstrated recent increases in wetness in streamflow in the Aspen Parkland ecoregion that form the northern Prairie against

the Boreal Forest.

In the Prairie pothole region, there is high inter-annual variability of annual precipitation (Millett et al 2008; Hayashi et al 2016), and there has been oscillation on the decadal scale between wet and dry conditions (Winter and Rosenberry 1998). van der Kamp et al (2008) examined a set of closed-basin lakes in the Prairie pothole region; long-term declining water

levels in these 16 basins over the 20th Century provides a measure of the balance between precipitation and evaporation. McKenna et al (2017) show that since the 1990's the southern Prairie pothole region has been influenced by an extended period of increased wetness\ resulting in higher water levels

Seager et al (2017a, 2017b) describe how the 100th meridian divides North America into an

arid west and a humid east that is expressed in vegetation, hydrology and agriculture. This gradient arises from atmospheric circulation and the transport of moisture. In winter, areas west of the meridian are sheltered from Pacific storm precipitation; in summer a southerly flow moves air from the southwest and Gulf of Mexico northwards with a strong west-east moisture transport. Seager et al (2017b) demonstrate that under increasing greenhouse gases the divide

is expected to move eastward, resulting in spread of aridity. The results presented here indicate increasing wetness near that meridian is shifting westward in Canada.

*Mountains*

Most of the basins (73%) in the Cordillera (Montane, Boreal, and Taiga) fall into Change Pattern 5 (Table 4, Figure S19) which has a general lack of structure in changes. Many of

these basins show a few periods with increases or decreases in flow for several periods consistent with freshet timing changes (e.g. Figure 1 and Figure 2), but there is a lack of consistency as indicated by the inability for a cluster of similar patterns to be formed (statistically). These timing shifts in have been widely reported (Leith and Whitfield 1998; Luckman 1998; Whitfield and Cannon 2000; Rood and Samuelson 2005; Forbes et al 2011;

Bennett et al 2015; Luce 2018; Philipsen et al 2018; and others). Basins in these ecozones do



have trends in NDVI, NDWI, and NDSI, but the results are mixed (Table 7). The fraction of basins with decreasing trends in mean NDVI that only just exceeds the threshold in the Montane Cordillera (0.06), both mean NDWI and mean NDSI exceed the threshold in the Boreal Cordillera (0.25) and Taiga Cordillera (1.00). Increases in mean NDVI exceed the

threshold in the Boreal Cordillera (0.25) and Taiga Cordillera (1.00) and there are no increasing trends in these three ecozones for either NDWI or NDSI suggest that changes are more prevalent in the north than in the south. Using a modelling approach, (Bennett et al 2012; Schnorbus et al 2014) demonstrated that detecting climate driven changes in basins in the British Columbia Rocky Mountains were difficult because of interannual variability.

Despite the ongoing deglaciation in the mountains of the west (Clarke et al 2015) basins in the Canadian Rockies can be resilient to change (Harder et al 2015, Whitfield and Pomeroy 2016).

### 3.4 Remaining Questions, Significance and Future Work

In the Arctic, the patterns, magnitude, and mechanisms of hydrological and ecological change are often unpredictable or difficult to separate from other drivers (Hinzman et al 2005). In many disciplines the complexity of the Arctic is challenged by a lack of scientific knowledge, observational and experimental time series, and the technical and logistic constraints of research in the Arctic. Ireson et al (2015) indicate that the current monitoring of climate,

hydrology, and ecology are insufficient for understanding the potential responses to both human activities and a changing climate. Even with inclusion of nearly 200 stations that are observed seasonally, there are regions in the study domain, particularly north of 60° that have too few observations and the basins sampled are large compared to the southern regions.

Climate signals, particularly for the Pacific Decadal Oscillation [PDO] and Arctic Oscillation

[AO] were not considered here; Thorne and Woo (2011) demonstrate that within mountainous areas, such as the Cordillera, the hydrological response to the regional climate variability signal is likely to be modified by local factors including location, topography, and land characteristics. Bennett et al (2015) showed some connections of streamflow in Boreal Alaska to PDO, but not to AO. Rood et al (2017) showed that the interannual variability of the Liard River was

correlated to the PDO. Alternate approaches will be required to properly assess any





relationships and would have to incorporate the multitude of station record lengths and partial year streamflow records.

Bennett et al (2015) suggest that broad-scale studies examining streamflow trends and timing changes should employ multiple methods across different scales and consider regime-

dependent shifts to better identify and understand changes.  Are landscape change related to initial state, and are they a response to or a driver of hydrological change?  Here, it was shown that there are coincident and overlapping regions of change in streamflow and in landscape. Further studies are required to determine drivers and responses.

A basic premise of this study was to let the available data tell the story of hydrological structure

and change in this study area.  Including data from seasonal stations that only report a part of the years nearly doubled the number of stations available for analysis, and most of these stations are located in the Prairie ecozone which has often been omitted in studies because of the lack of continuous data.  The available data are messy, and there is an uneven distribution of stations by ecozones.  The ability to determine the magnitudes of annual trends was lost in the approach

used, but by including the data from seasonal stations, the results provide an interesting spatial story.  There are timing shifts in many Cordillera basins, but these appear resilient to change. In the north, winter streamflows are increasing, and there are changes detected in landscape indices.  The western Boreal Plains exhibit decreasing trends in streamflow and basins are less green.  There is a complex pattern of increased streamflow across the Prairies with more

frequent increasing in streamflow in the eastern Prairies, particularly in the Aspen Parkland ecoregion.

The results presented here are informing several science research agendas in Canada and internationally.  The motivation for this study came from the NSERC Changing Cold Regions Network study of 2013-2018 (DeBeer et al., 2015) which had an objective to integrate existing

and new sources of data with improved predictive and observational tools to understand, diagnose and predict interactions amongst the cryospheric, ecological, hydrological, and climatic components of the changing Earth system at multiple scales, with a geographical focus on Western Canada's rapidly changing cold interior.  The results also contribute to the Global Water Futures programme of 2016-2023 (www.globalwaterfutures.ca), which has an

overarching goal to deliver risk management solutions to manage water futures in Canada.  The first step to managing future water changes is to understand those of the recent past that are



currently underway and this study makes a strong contribution to that process. In particular, this study represents a step forward in addressing the complexity of change; there are many studies of individual basins where, when results are considered individually, tend to be more

anecdotal than systematic. It is indeed simpler and easier when there are only a few cases to consider with common variables and record length, as in many studies, but with large numbers of basins the tools for dealing with significant changes are more limited. Sensitivity studies that assess the limits of partial year analysis of hydrological structure and change are required and a logical next step.


**Conclusions**

This study uses accepted techniques with a very large dataset to integrate existing and new sources of data to understand and diagnose interactions amongst the climatic, hydrological, and ecological components of Western Canada's rapidly changing cold interior. Methods were

used in a novel manner that treated hydrometric stations with only warm season observations in the same way as continuously observed basins. A new clustering methodology, dynamic time warping, was used to separate stations based on Streamflow Regime type. A clustering of the seasonal pattern of Streamflow Regime change allowed the examination of the relationship of change to Streamflow Regime types and to the spatial domain. Spatial location

rather than Streamflow Regime type is a stronger determinant of change and is consistent with large-scale change in the climate system.

Trends in indices obtained from time series of indices derived from Landsat observations allow the changes in the landscape of basins to be related to hydrological changes. While there are not simple one-to-one correspondences among Streamflow Regime types, seasonal pattern

changes, and landscape indices, four prominent regions of changes were diagnosed; these regions were also identified by Rodell et al (2018) to have changes in water storage as determined from GRACE satellites.

*The Mackenzie Basin*

Increased streamflows, particularly in winter, are taking place in the northern portion of the

Mackenzie Basin. In these basins, and in the general region, mean NDVI has not changed, but



mean NDWI and NDSI have decreased. Degrading permafrost resulting in increased winter streamflow is an important change, which has been observed to take place in this region, and which may be driving the observed changes in NDWI. Decreasing snowcover, evidenced by decreased NDSI may be reflected in the shift in the partition of rainfall and snowfall.

*The western Boreal Plains and the western Prairies*

Decreased streamflows occur in these basins. In this area in general, mean NDVI has significantly decreased, but along the western margin mean NDWI and NDSI have increased. Decreased NDVI is consistent with decreased streamflow; as basins become drier vegetation is impacted. Increases in mean NDWI and NDSI occur only at the western margin of this
region suggesting a zone with complex gradients and changes.

*The Prairies*

Increased streamflows in summer and fall are occurring across the Prairies, particularly in the east, and along the northern margin and the southern margin of the Boreal Plains. Three types of hydrological changes are evident with differing spatial locations, but the overlap between
types suggests that there are multiple processes involved. In these basins, and in this area in general, mean NDVI has significantly increased, mean NDWI has increased, and mean NDSI has not changed. Basins in the eastern Prairie show increased streamflows through the entire period of the year that data are available. Some basins show increases only during summer and fall and these are centred in Saskatchewan. These increases are indicative of changes in the
Streamflow Regime from nival to pluvial. Some basins show only a narrow time with increased streamflow, which is the largest of these three change patterns, and are found in the western Prairie and along the margin between the Prairie and Boreal Plains. In these basins, mean NDVI has not changed, but NDWI has increased in some basins in Saskatchewan and Alberta, and mean NDSI has increased in Saskatchewan. These differences may reflect the differences
in the three change patterns. The existence of the east to west gradient of these changes was predicted by previous climatological studies (Borchert 1950; Rosenberg 1987; Luckman 1990).

*The Cordillera*

Mountain basins appear to be resilient to change. These basins demonstrate several hydrograph types but generally lack structure in trend patterns. Individually, these basin do



show periods with increases or decreases in streamflow consistent with freshet timing

changes, as has been reported elsewhere, but there is sufficient inconsistency among the

basins to define a specific pattern. Basins in these ecozones do have trends in NDVI, NDWI,

and NDSI, but the results are mixed (Table 7). There are decreasing trends in mean NDVI in

the Montane Cordillera, and mean NDWI and mean NDSI in the Boreal Cordillera and Taiga

Cordillera, and increasing trends in mean NDVI in the Boreal Cordillera and Taiga Cordillera

and there are no increasing trends in these three ecozones for either NDWI or NDSI;

decreasing trends are more prevalent in the north than in the south.

## 4        Code Availability


## 5        Data Availability

Streamflow data are available from Water Survey of Canada (Environment and Climate

Change Canada). Satellite imagery is available from NASA, USGS, and NOAA through

Google Earth Engine.

## 6        Appendices

Supplementary Figures

## 7        Author Contributions

PHW and JWP outlined the original study form with input from KRS. PK and PHW designed

the landscape indices extracted that was performed by PK. The statistical analysis of





streamflow was conducted by PHW with assistance from KRS. The trend analysis of landscape

indices was conducted by PHW with input from PK and JWP. The manuscript was drafted by

PHW; all authors contributed to the interpretation and final manuscript.

## 8 Competing Interests

The authors declare that they have no conflict of interest.

## 9 Acknowledgements

Funding was provided by the Natural Science and Engineering Research Council of Canada

through Discovery Grants and through the Changing Cold Regions Network, and by the

Canada Research Chairs, the Canada Excellence Research Chairs programs and the Global

Water Futures program. Streamflow data were obtained from Water Survey of Canada

(Environment and Climate Change Canada). Satellite imagery was provided by NASA,

USGS, and NOAA through Google Earth Engine. We appreciate being able to use the R-

packages identified in the methods and the contributions of many people to the

`CSHShydRology` package and to the R Development Core Team. The comments and

suggestions of the Editors and Reviewers of this Special Issue are appreciated. Finally, we

appreciate the enthusiastic support of colleagues in the members of the Changing Cold

Regions Network who have commented and made suggestions throughout this study.

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



Table 1. Hydrometric stations included in the analysis. Only stations that are in these three basins were considered; to be included they needed to be designated as having natural streamflow, active, and have more than 30 years of record. The other hydrometric were associated with a Changing Cold Regions Network Water, 1355 Ecosystem, Cryosphere and Climate (WECC) Observatory but have less than 30 years of streamflow records.

| Drainage Basin | Water Survey Drainage Basin Code | Hydrometric Stations # of sites | Other Hydrometric stations |
|---|---|---|---|
| South Saskatchewan | 05 | 232 | 05BF016 05ME007 |
| North Saskatchewan | 07 | 121 | |
| Mackenzie | 10 | 38 | 10ED009 |
| | | 392 | 3 |





Table 2. Streamflow Regime Type classification in relation to the ecozone in which the station is located.

| Ecozone | Streamflow Regime Type | | | | | | | | | | | | |
| | 1 | 2 | 3 | 4 | 5 | 6 | 7 | 8 | 9 | 10 | 11 | 12 | n |
|---|---|---|---|---|---|---|---|---|---|---|---|---|---|
| Southern Arctic | | 1 | | | | | | | | | | | 1 |
| Taiga Plains | 5 | 3 | | | 9 | | | | 1 | | 5 | 1 | 24 |
| Taiga Shield | 1 | | | 2 | 1 | | | | 3 | | | | 7 |
| Boreal Shield | 13 | | 2 | 2 | 6 | | | | 1 | | 3 | | 27 |
| Boreal Plains | 31 | 28 | 17 | 1 | 81 | 1 | 1 | | | | | | 160 |
| Prairies | 11 | 53 | 3 | | 49 | | | 1 | | 2 | | | 119 |
| Montane Cordillera | 31 | | | 5 | 2 | | | | | | | | 38 |
| Boreal Cordillera | 11 | | | | | | | | | | | 2 | 13 |
| Taiga Cordillera | 1 | | | | | | | | | | 1 | 1 | 3 |
| Hudson Plains | | | | | | | | | | | 3 | | 3 |
| n | 104 | 85 | 22 | 10 | 148 | 1 | 1 | 1 | 5 | 2 | 12 | 4 | 395 |






Table 3. Trend patterns for 395 stations against their Streamflow Regime Type classification.

| Streamflow Regime Type | Trend Pattern | | | | | | |
| | 1 | 2 | 3 | 4 | 5 | 6 | n |
|---|---|---|---|---|---|---|---|
| 1 | 1 | 2 | 5 | 13 | 75 | 8 | 104 |
| 2 | 9 | 10 | 10 | 16 | 39 | 1 | 85 |
| 3 | | | 2 | 2 | 17 | 1 | 22 |
| 4 | | | | 1 | 9 | | 10 |
| 5 | 6 | 10 | 13 | 17 | 101 | 1 | 148 |
| 6 | | | | 1 | | | 1 |
| 7 | | | | | 1 | | 1 |
| 8 | | | 1 | | | | 1 |
| 9 | 1 | | 1 | | 3 | | 5 |
| 10 | 2 | | | | | | 2 |
| 11 | | | | | 8 | 4 | 12 |
| 12 | | | | | 1 | 3 | 4 |
| n | 19 | 22 | 32 | 50 | 254 | 18 | 395 |





Table 4. Trend Patterns in relation to the ecozone in which the station is located.

|  | Trend Pattern | | | | | | |
| --- | --- | --- | --- | --- | --- | --- | --- |
| **Ecozone** | **1** | **2** | **3** | **4** | **5** | **6** | *n* |
| Southern Arctic |  |  |  |  |  | 1 | *1* |
| Taiga Plains | 1 | 2 |  | 1 | 13 | 7 | *24* |
| Taiga Shield | 1 |  | 1 |  | 4 | 1 | *7* |
| Boreal Shield |  | 1 | 1 | 3 | 21 | 1 | *27* |
| Boreal Plains | 3 | 9 | 21 | 13 | 114 |  | *160* |
| Prairies | 14 | 10 | 7 | 29 | 58 | 1 | *119* |
| Montane Cordillera |  |  | 2 | 4 | 32 |  | *38* |
| Boreal Cordillera |  |  |  |  | 9 | 4 | *13* |
| Taiga Cordillera |  |  |  |  |  | 3 | *3* |
| Hudson Plains |  |  |  |  | 3 |  | *3* |
| *n* | *19* | *22* | *32* | *50* | *254* | *18* | *395* |





Table 5. Trends in landscape and hydrological storage indices in relation to Streamflow Regime Type. The numbers are the fraction of stations showing a trend; values greater than 0.05 are in bold.

| All Changes | Stations | NDVI max | NDVI mean | NDVI min | NDWI max | NDWI mean | NDWI min | NDSI max | NDSI mean | NDSI min |
|---|---|---|---|---|---|---|---|---|---|---|
| 1 | 99 | **0.15** | **0.09** | **0.10** | 0.03 | **0.07** | **0.15** | 0.03 | **0.07** | **0.16** |
| 2 | 79 | **0.22** | 0.03 | **0.18** | **0.05** | **0.14** | **0.15** | **0.06** | 0.04 | **0.06** |
| 3 | 21 | **0.14** | 0.05 | **0.10** | 0.05 | **0.10** | **0.29** | 0.05 | 0.00 | **0.29** |
| 4 | 10 | 0.00 | 0.00 | **0.30** | 0.00 | **0.10** | **0.40** | 0.00 | **0.10** | **0.20** |
| 5 | 142 | **0.19** | 0.04 | **0.11** | **0.06** | **0.10** | **0.18** | **0.08** | 0.03 | **0.09** |
| 6 | 1 | 0.00 | 0.00 | 0.00 | 0.00 | 0.00 | 0.00 | 0.00 | 0.00 | 0.00 |
| 7 | 1 | 0.00 | 0.00 | 0.00 | 0.00 | 0.00 | 0.00 | 0.00 | 0.00 | 0.00 |
| 8 | 0 | | | | | | | | | |
| 9 | 5 | **0.20** | **0.40** | 0.00 | **0.20** | **0.40** | **0.20** | **0.20** | **0.40** | **0.20** |
| 10 | 2 | **0.50** | 0.00 | 0.00 | 0.00 | 0.00 | 0.00 | 0.00 | 0.00 | **0.50** |
| 11 | 12 | 0.00 | **0.42** | **0.25** | **0.25** | **0.42** | **0.25** | **0.17** | **0.42** | **0.25** |
| 12 | 3 | 0.00 | **0.67** | **0.33** | **0.33** | **0.33** | **0.33** | **0.33** | **0.33** | **0.33** |
| *Decreases* | | | | | | | | | | |
| 1 | 99 | **0.05** | 0.03 | 0.04 | 0.02 | **0.05** | **0.08** | 0.02 | **0.07** | **0.08** |
| 2 | 79 | 0.03 | 0.01 | **0.18** | 0.00 | 0.00 | 0.00 | 0.00 | 0.00 | 0.01 |
| 3 | 21 | **0.14** | 0.00 | 0.05 | 0.00 | **0.05** | **0.14** | 0.00 | 0.00 | **0.10** |
| 4 | 10 | 0.00 | 0.00 | **0.10** | 0.00 | **0.10** | **0.30** | 0.00 | **0.10** | **0.10** |
| 5 | 142 | **0.13** | 0.02 | **0.09** | 0.01 | 0.01 | **0.09** | 0.01 | 0.00 | 0.05 |
| 6 | 1 | 0.00 | 0.00 | 0.00 | 0.00 | 0.00 | 0.00 | 0.00 | 0.00 | 0.00 |
| 7 | 1 | 0.00 | 0.00 | 0.00 | 0.00 | 0.00 | 0.00 | 0.00 | 0.00 | 0.00 |
| 8 | 0 | | | | | | | | | |
| 9 | 5 | **0.20** | 0.00 | 0.00 | **0.20** | **0.40** | **0.20** | **0.20** | **0.40** | **0.20** |
| 10 | 2 | **0.50** | 0.00 | 0.00 | 0.00 | 0.00 | 0.00 | 0.00 | 0.00 | 0.00 |
| 11 | 12 | 0.00 | 0.00 | 0.00 | **0.25** | **0.42** | **0.08** | **0.17** | **0.42** | **0.08** |
| 12 | 3 | 0.00 | 0.00 | 0.00 | **0.33** | **0.33** | 0.00 | **0.33** | **0.33** | 0.00 |
| *Increases* | | | | | | | | | | |
| 1 | 99 | **0.10** | **0.06** | **0.06** | 0.01 | 0.02 | **0.07** | 0.01 | 0.00 | **0.08** |
| 2 | 79 | **0.19** | 0.01 | 0.00 | **0.05** | **0.14** | **0.15** | **0.06** | 0.04 | **0.05** |
| 3 | 21 | 0.00 | 0.05 | 0.05 | **0.05** | **0.05** | **0.14** | 0.05 | 0.00 | **0.19** |
| 4 | 10 | 0.00 | 0.00 | **0.20** | 0.00 | 0.00 | **0.10** | 0.00 | 0.00 | **0.10** |
| 5 | 142 | **0.06** | 0.01 | 0.01 | **0.05** | **0.08** | **0.09** | **0.07** | 0.03 | 0.04 |
| 6 | 1 | 0.00 | 0.00 | 0.00 | 0.00 | 0.00 | 0.00 | 0.00 | 0.00 | 0.00 |
| 7 | 1 | 0.00 | 0.00 | 0.00 | 0.00 | 0.00 | 0.00 | 0.00 | 0.00 | 0.00 |
| 8 | 0 | | | | | | | | | |
| 9 | 5 | 0.00 | **0.40** | 0.00 | 0.00 | 0.00 | 0.00 | 0.00 | 0.00 | 0.00 |
| 10 | 2 | 0.00 | 0.00 | 0.00 | 0.00 | 0.00 | 0.00 | 0.00 | 0.00 | **0.50** |





| 11 | 12 | 0.00 | 0.42 | 0.25 | 0.00 | 0.00 | 0.17 | 0.00 | 0.00 | 0.17 |
| 12 | 3 | 0.00 | 0.67 | 0.33 | 0.00 | 0.00 | 0.33 | 0.00 | 0.00 | 0.33 |
| | 375 | | | | | | | | | |






Table 6. Changes in landscape and hydrological storage indices by Trend Pattern. The numbers are the fraction of stations showing a trend; values greater than 0.05 are in bold.

| *All changes* | *Stations* | NDVI max | NDVI mean | NDVI min | NDWI max | NDWI mean | NDWI min | NDSI max | NDSI mean | NDSI min |
|---|---|---|---|---|---|---|---|---|---|---|
| 1 | *17* | **0.18** | **0.12** | **0.29** | **0.12** | **0.18** | **0.18** | **0.12** | **0.06** | **0.18** |
| 2 | *22* | **0.36** | 0.00 | **0.23** | 0.05 | **0.14** | **0.09** | **0.14** | **0.09** | **0.09** |
| 3 | *30* | 0.00 | 0.00 | **0.07** | **0.10** | **0.07** | 0.03 | **0.13** | 0.03 | 0.03 |
| 4 | *47* | **0.36** | **0.09** | **0.15** | **0.09** | **0.15** | **0.21** | **0.09** | **0.06** | 0.04 |
| 5 | *243* | **0.14** | **0.05** | **0.10** | 0.05 | **0.09** | **0.21** | 0.04 | 0.04 | **0.15** |
| 6 | *16* | **0.06** | **0.44** | **0.25** | **0.06** | **0.38** | **0.06** | **0.06** | **0.38** | **0.25** |
| *Decreases* | | | | | | | | | | |
| 1 | *17* | **0.12** | 0.00 | **0.29** | **0.12** | **0.06** | 0.00 | **0.12** | **0.06** | 0.00 |
| 2 | *22* | **0.14** | 0.00 | **0.23** | 0.00 | 0.00 | 0.05 | 0.00 | 0.05 | 0.05 |
| 3 | *30* | 0.00 | 0.00 | **0.07** | 0.00 | 0.00 | 0.00 | 0.00 | 0.00 | 0.00 |
| 4 | *47* | **0.06** | 0.00 | **0.09** | 0.02 | 0.02 | 0.02 | 0.02 | 0.02 | 0.04 |
| 5 | *243* | **0.09** | 0.03 | **0.07** | 0.02 | 0.04 | **0.11** | 0.01 | 0.03 | **0.07** |
| 6 | *16* | 0.00 | 0.00 | **0.06** | **0.06** | **0.38** | 0.00 | **0.06** | **0.38** | 0.00 |
| *Increases* | | | | | | | | | | |
| 1 | *17* | **0.06** | **0.12** | 0.00 | 0.00 | **0.12** | **0.18** | 0.00 | 0.00 | **0.18** |
| 2 | *22* | **0.23** | 0.00 | 0.00 | 0.05 | **0.14** | 0.05 | **0.14** | 0.05 | 0.05 |
| 3 | *30* | 0.00 | 0.00 | 0.00 | **0.10** | **0.07** | 0.03 | **0.13** | 0.03 | 0.03 |
| 4 | *47* | **0.30** | **0.09** | **0.06** | **0.06** | **0.13** | **0.19** | **0.06** | 0.04 | 0.00 |
| 5 | *243* | **0.05** | 0.02 | 0.04 | 0.02 | **0.05** | **0.10** | 0.03 | 0.01 | **0.07** |
| 6 | *16* | **0.06** | **0.44** | **0.19** | 0.00 | 0.00 | **0.06** | 0.00 | 0.00 | **0.25** |
| | *375* | | | | | | | | | |





Table 7. Changes in landscape and hydrological storage indices in relation to Ecozones.  The numbers are the fraction of stations showing a trend; values greater than 0.05 are in bold.

| All changes | Stations | NDVI max | NDVI mean | NDVI min | NDWI max | NDWI mean | NDWI min | NDSI max | NDSI mean | NDSI min |
|---|---|---|---|---|---|---|---|---|---|---|
| Southern Arctic | 1 | 1.00 | 0.00 | 1.00 | 0.00 | 0.00 | 0.00 | 0.00 | 0.00 | 0.00 |
| Taiga Plains | 20 | 0.20 | 0.25 | 0.20 | 0.15 | 0.25 | 0.15 | 0.15 | 0.25 | 0.30 |
| Taiga Shield | 6 | 0.00 | 0.17 | 0.17 | 0.17 | 0.17 | 0.17 | 0.17 | 0.17 | 0.00 |
| Boreal Shield | 25 | 0.28 | 0.20 | 0.20 | 0.04 | 0.08 | 0.44 | 0.00 | 0.12 | 0.28 |
| Boreal Plains | 157 | 0.12 | 0.04 | 0.08 | 0.05 | 0.08 | 0.18 | 0.06 | 0.02 | 0.11 |
| Prairies | 112 | 0.29 | 0.02 | 0.16 | 0.06 | 0.13 | 0.14 | 0.08 | 0.04 | 0.07 |
| Montane Cordillera | 36 | 0.03 | 0.06 | 0.08 | 0.00 | 0.06 | 0.14 | 0.00 | 0.03 | 0.08 |
| Boreal Cordillera | 12 | 0.00 | 0.17 | 0.17 | 0.17 | 0.25 | 0.17 | 0.17 | 0.25 | 0.42 |
| Taiga Cordillera | 3 | 0.00 | 1.00 | 0.33 | 0.00 | 1.00 | 0.00 | 0.00 | 1.00 | 0.33 |
| Hudson Plains | 3 | 0.00 | 0.00 | 0.00 | 0.00 | 0.00 | 0.33 | 0.00 | 0.00 | 0.00 |
| **Decreases** | | | | | | | | | | |
| Southern Arctic | 1 | 0.00 | 0.00 | 1.00 | 0.00 | 0.00 | 0.00 | 0.00 | 0.00 | 0.00 |
| Taiga Plains | 20 | 0.20 | 0.00 | 0.10 | 0.15 | 0.25 | 0.05 | 0.15 | 0.25 | 0.10 |
| Taiga Shield | 6 | 0.00 | 0.00 | 0.17 | 0.17 | 0.17 | 0.17 | 0.17 | 0.17 | 0.00 |
| Boreal Shield | 25 | 0.08 | 0.00 | 0.00 | 0.04 | 0.04 | 0.20 | 0.00 | 0.12 | 0.16 |
| Boreal Plains | 157 | 0.10 | 0.03 | 0.07 | 0.01 | 0.02 | 0.11 | 0.01 | 0.00 | 0.06 |
| Prairies | 112 | 0.06 | 0.01 | 0.15 | 0.00 | 0.00 | 0.01 | 0.00 | 0.00 | 0.02 |
| Montane Cordillera | 36 | 0.03 | 0.06 | 0.03 | 0.00 | 0.03 | 0.08 | 0.00 | 0.03 | 0.08 |
| Boreal Cordillera | 12 | 0.00 | 0.00 | 0.00 | 0.17 | 0.25 | 0.00 | 0.17 | 0.25 | 0.00 |
| Taiga Cordillera | 3 | 0.00 | 0.00 | 0.00 | 0.00 | 1.00 | 0.00 | 0.00 | 1.00 | 0.00 |
| Hudson Plains | 3 | 0.00 | 0.00 | 0.00 | 0.00 | 0.00 | 0.33 | 0.00 | 0.00 | 0.00 |
| **Increases** | | | | | | | | | | |
| Southern Arctic | 1 | 1.00 | 0.00 | 0.00 | 0.00 | 0.00 | 0.00 | 0.00 | 0.00 | 0.00 |
| Taiga Plains | 20 | 0.00 | 0.25 | 0.10 | 0.00 | 0.00 | 0.10 | 0.00 | 0.00 | 0.20 |
| Taiga Shield | 6 | 0.00 | 0.17 | 0.00 | 0.00 | 0.00 | 0.00 | 0.00 | 0.00 | 0.00 |
| Boreal Shield | 25 | 0.20 | 0.20 | 0.20 | 0.00 | 0.04 | 0.24 | 0.00 | 0.00 | 0.12 |
| Boreal Plains | 157 | 0.02 | 0.01 | 0.01 | 0.04 | 0.06 | 0.08 | 0.05 | 0.02 | 0.05 |
| Prairies | 112 | 0.22 | 0.01 | 0.01 | 0.06 | 0.13 | 0.13 | 0.08 | 0.04 | 0.05 |
| Montane Cordillera | 36 | 0.00 | 0.00 | 0.06 | 0.00 | 0.03 | 0.06 | 0.00 | 0.00 | 0.00 |





| | | | | | | | | | |
|---|---|---|---|---|---|---|---|---|---|
| Boreal Cordillera | *12* | 0.00 | **0.17** | **0.17** | 0.00 | 0.00 | **0.17** | 0.00 | 0.00 | **0.42** |
| Taiga Cordillera | *3* | 0.00 | **1.00** | **0.33** | 0.00 | 0.00 | 0.00 | 0.00 | 0.00 | **0.33** |
| Hudson Plains | *3* | 0.00 | 0.00 | 0.00 | 0.00 | 0.00 | 0.00 | 0.00 | 0.00 | 0.00 |
| | *375* | | | | | | | | | |





Table 8. Summary of recession slopes amongst the twelve Streamflow Regime Types. Units are z-score/length
estimated from Figure 11. Types with * have only one member and are excluded here.

| Streamflow Regime Type | Recession slope 1 | Recession slope 2 | Recession slope 3 |
|---|---|---|---|
| 1 | -0.22 | -0.04 | NA |
| 2 | Nonlinear | | |
| 3 | -0.22 | -0.04 | |
| 4 | -0.16 | -0.06 | |
| 5 | -0.22 | -0.06 | 0.00 |
| 6* | | | |
| 7* | | | |
| 8* | | | |
| 9 | -0.04 | NA | NA |
| 10 | -0.06 | NA | NA |
| 11 | -0.22 | 0.00 | -0.04 |
| 12 | -0.06 | NA | NA |

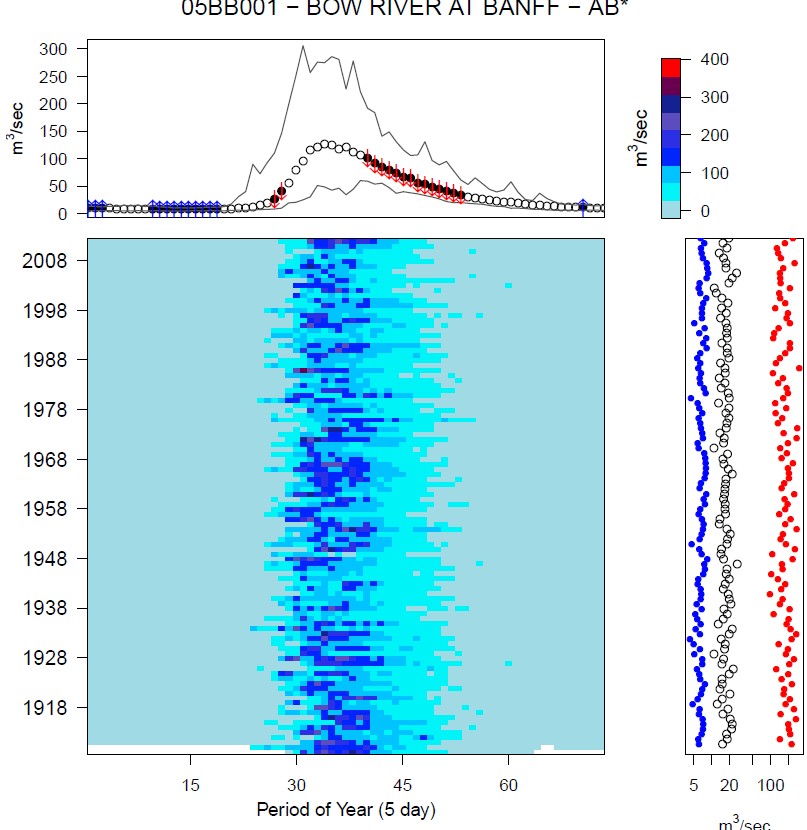


Figure 1. Plot of observed flows in the Reference Hydrologic Basin station 05BB001 Bow River at Banff, Alberta. The main panel shows the five day periods of the year against the years of record. White space indicates missing observations and the colours represent flow magnitudes scaled according to the bar in the upper right corner. The upper panel show the maximum, median and minimum flow for each of the five day periods and red (blue) arrows

indicate statistically significant decreases (increases) using Mann-Kendall τ a p≤0.05. The panel on the right shows the annual minima, median (open circle), and maxima; statistically significant decreasing (increasing) trends (Mann-Kendall τ at p≤0.05) are indicated by red (blue). Whenever the station is a member of the reference hydrologic basin network (RHBN) an * appears at the end of the station name.






Figure 2. Plot of observed flows in the 05BA001 Bow River at Lake Louise, Alberta; a natural flow station. The main panel shows the five day periods of the year against the years of record. White space indicates missing observations and the colours are scales according to the bar in the upper right corner. The upper panel show the maximum, median and minimum flow for each of the five day periods and red (blue) arrows indicate statistically significant decreases (increases) using Mann-Kendall τ at p≤0.05. The panel on the right shows the annual minima, median (open circle), and maxima; statistically significant decreasing (increasing) trends (Mann-Kendall τ a p≤0.05) are indicated by red (blue).






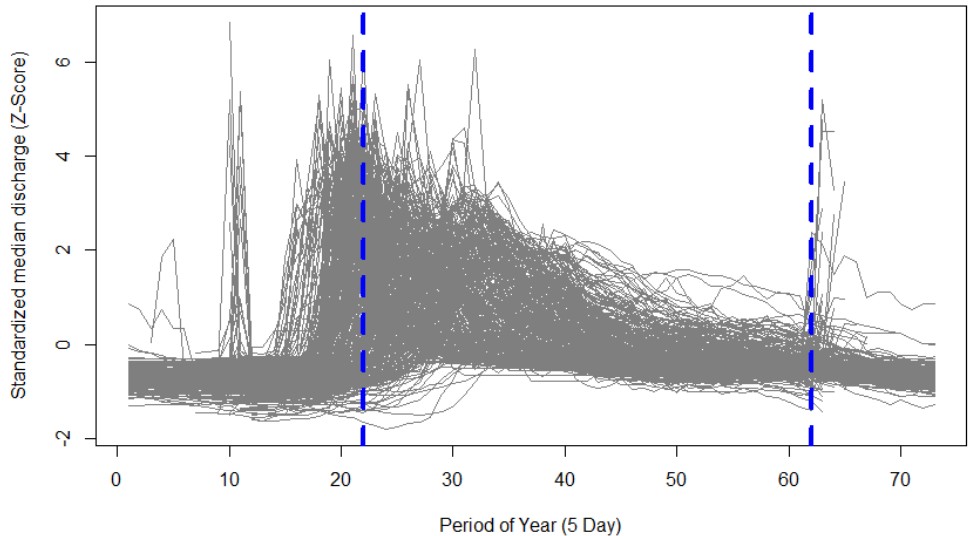

Figure 3. Standardized median streamflow for 5-day periods from the 395 hydrometric stations. The flow from each station was standardized removing the mean and dividing by the standard deviation. Only the period between 1 April and 30 September (indicated by the region between the blue dashed lines) is considered, as seasonally 1410 operated stations have no observations during winter months.





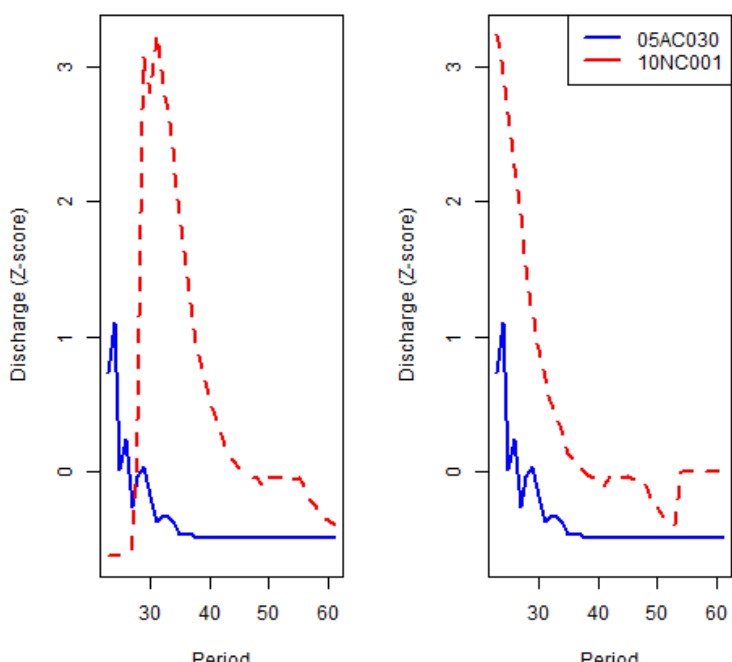

Figure 4. Example of alignment of two time series using dynamic time warping [dtw] from stations 05AC030
Snake Creek near Vulcan AB and 10NC001 Anderson River below Carnwath River NT.




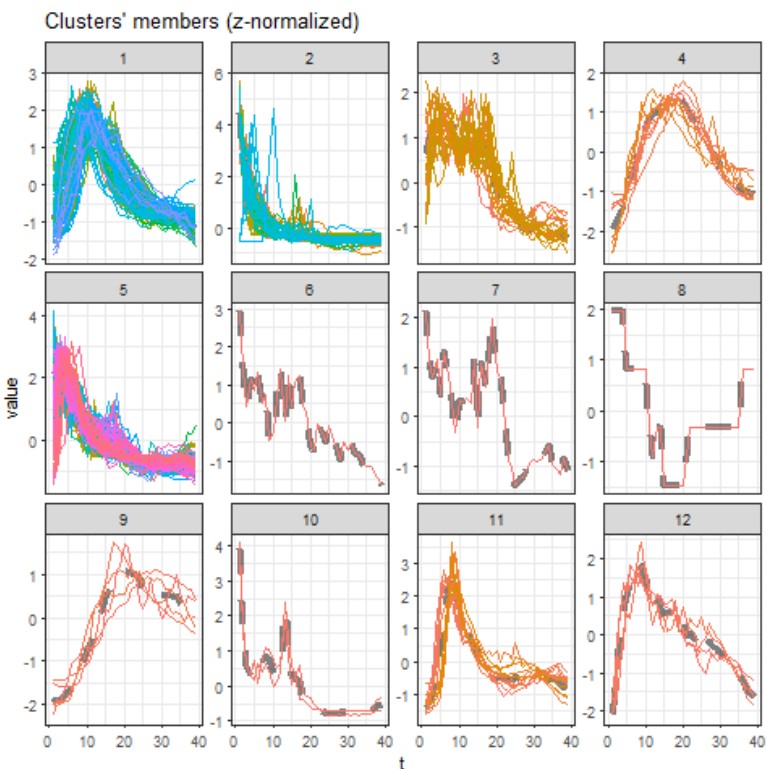

Figure 5. Streamflow Regime Types produced by clustering of the 395 standardized median 5 day streamflows using dynamic time warping. The heavy dashed line is the centroid of the cluster. Note that the number on the x axis is for the aligned series (1-39) as opposed to 23-61. The y axis value is Z-score.




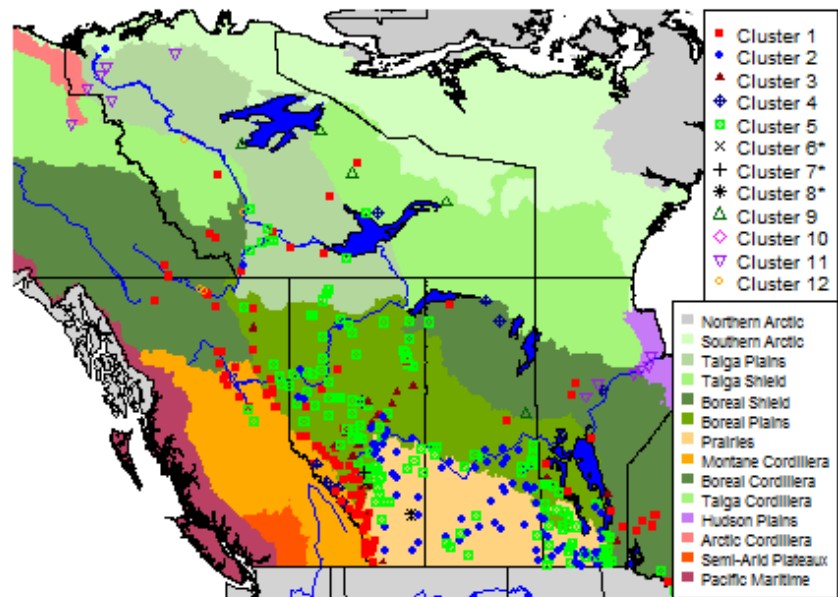

Figure 6. Locations of the 12 Streamflow Regime Types from the changing cold regions domain overlain on the ecosystems of western Canada. Clusters marked with an * have only a single member.






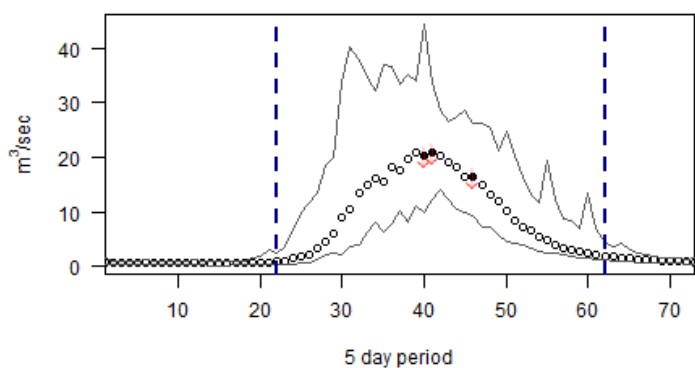

Figure 7. Example of the trend portion of the summary hydrograph, the two dashed lines indicate the start and end of the common window from period 23 to period 61.




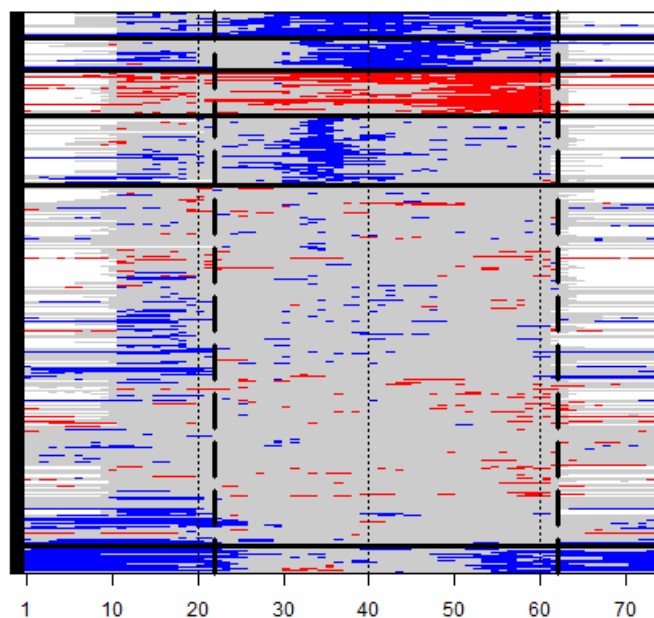

Figure 8. Trend Patterns in the 395 stations, significant increases (blue) and significant decreases (red), no trend (gray) and missing (white). The stations are ordered by Trends Pattern (cluster) number and stationID. Data outside the dashed lines was not used in the clustering, but is shown where available (periods 1-22 and 62-73).


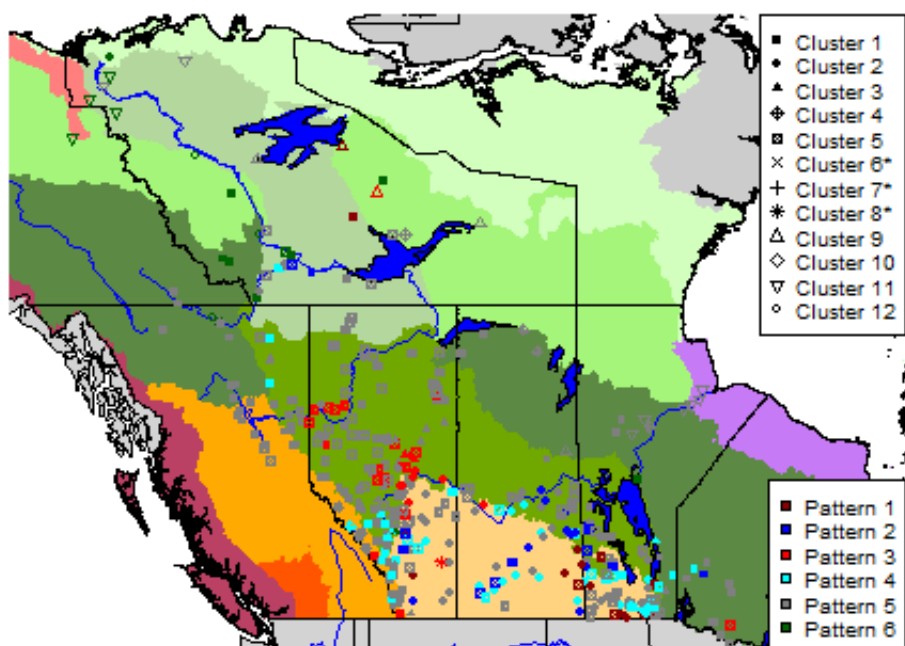

Figure 9. Trend Patterns (colour) and Streamflow Regime Types (cluster # with symbols) of the 395 stations in the study domain. The ecozone legend is as in Figure 6.

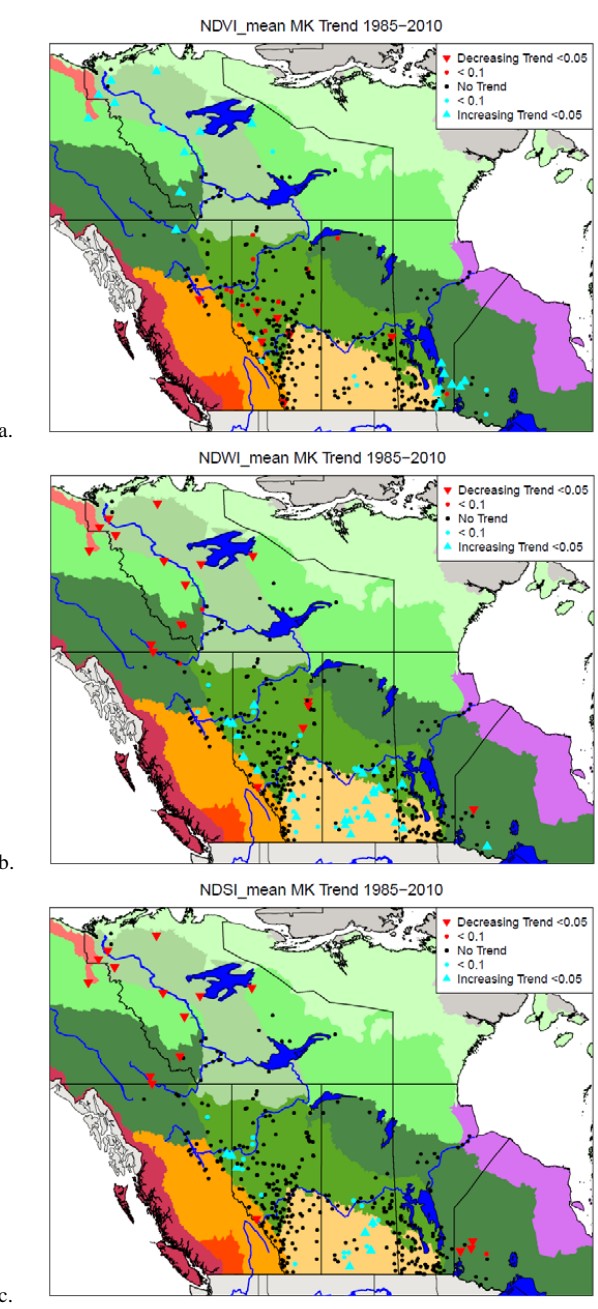

Figure 10. Trends in mean (a) NDVI, (b) NDWI, and (c) NDSI between 1985 and 2012. The ecozone legend is as
        in Figure 6.

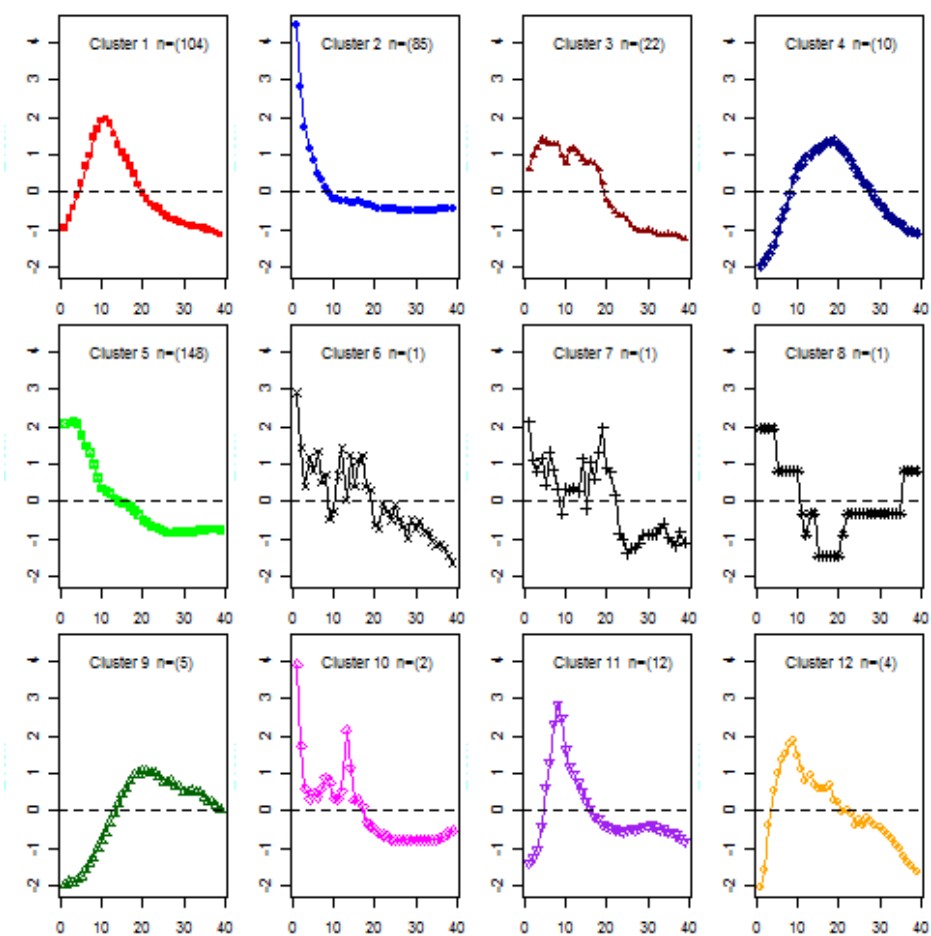

Figure 11. Centroids of the 12 Streamflow Regime Type clusters.
