# Peer review of "The Spatial Extent of Hydrological and Landscape Changes across the Mountains and Prairies of the Saskatchewan and Mackenzie Basins"

_Hydrology and Earth System Sciences, 2019_

## Referee Comment (RC1) · Anonymous Referee #1 · 27 Feb 2020

General comments:

This is a really interesting, innovative, and valuable study that deserves to be published in HESS. The existing database of streamflow measurements in western Canada east of the continental divide is expanded, perhaps greatly, compared to earlier statistical studies of hydrologic change in the region by adding datasets from streamgages that are only operated seasonally (i.e. not in winter). Some intriguing new analytical methods are introduced, in particular dynamic time warping. Further, Landsat imagery is interpreted alongside the statistical hydrology results, giving additional insights. The region itself also warrants close study in light of its size and, especially in its north-

ern reaches, relative lack of study compared to many other places. Unfortunately, the manuscript doesn't seem to do a great job of communicating the work, and it has the feeling of being only half-finished, as explained in some detail in the specific comments below. I therefore recommended publication of this (potentially excellent) article pending major revisions that entail among other things an extensive ground-up rewrite.

Specific comments:

The abstract feels clunky. There are unnecessary details (e.g. clarifying that the Landsat imagery analysis covers a different timeframe from the streamflow data analysis) while at the same time some of the big punch lines from the study seem to be missing. There are odd writing choices (such as using both quotation marks and capitalization for "Streamflow Regime Types" and "Trend Patterns" when neither of these terms/concepts is new in any way). And some of it is just plain confusing.

The use of data from rivers that are intermittent or ephemeral, because for example they're frozen completely or unmonitored in winter, is one of the more interesting and new aspects of this study. However, this dimension of the paper also feels like it hasn't been explored as much as it could and maybe should have. On the one hand, there may be a bit of missed opportunity here. The study doesn't seem to give much information on the main advantage of this approach, specifically, that it could hugely increase the sample size by pooling results across both perennial and intermittent/seasonal rivers; I don't see any numbers in the article about how much the available dataset is expanded by using this philosophy? And one of the most interesting questions that could be addressed in a study applying statistical change analysis to streamflow data from seasonal cold-regions rivers would be whether there have been changes in the annual no-flow duration and/or its dates of onset, etc; that hasn't been tackled here at all as far as I can see. At the same time, the approach used (if I understand it correctly) has the disadvantage of looking only at the common seasonal period between each streamgage time series, which means that wintertime flow measurements are not considered; that's the tradeoff. While some of the rivers in the study don't flow in winter,

many do. Given that some of the stronger changes in climate and hydrology at high latitudes and cold regions generally are being experienced in winter, certain important phenomena could be missed using this approach. That fact doesn't necessarily reduce the "publishability" of the paper, but it does imply that a clearer statement of scope and purpose ought to be made. Maybe I've just missed it (a real possibility given the clunky writing and organization of the paper) but I don't see anywhere in the title or abstract that the study is limited to the warm season and it doesn't look like there's much discussion in the paper as a whole of what this limitation means.

Lines 100-103, phrases like "by integrating different forms of data, which previously had only been treated separately and individually" and "By linking continuous and partial year data from a large number of hydrometric stations using only warm season data three important questions are addressed. . ." I worry that many readers might feel that simply using data from summer, or looking at Landsat data alongside the streamflow data analysis, doesn't really count as "data integration," which might be taken to mean something more ambitious than what was done here. It might be safer to simply say that an unusually large and diverse dataset was considered.

Not clear what lines 134-135 are intended to mean; typographical error?

The idea of not using a common period of record across all the streamgages runs contrary to almost all work in this field. That doesn't necessarily mean it's wrong, but it does seems likely to be viewed as a problem by some readers, and the subject probably requires more attention than it's been given in the article. For example, the authors write on lines 138-144 that "Because the periods of record, rather than a common period, are used it is not possible to compare the magnitudes of trends among the stations. Instead, the analyses are restricted to determining the existence of significant trends in individual five-day periods in (five-day) periods 23 to 61 (of 73), as shown in Figure 1 and 2." In addition to being a clunky and hard-to-read sentence, its logic seems doubtful. If the magnitudes of trends are not comparable between stations, why would the existences of trends be? One problem is statistical. Determining the existence of

trend using a statistical significance test (as done here) simply amounts to a measure of the magnitude of the observed trend relative to what you'd expect to see purely by chance; the concepts of magnitude and existence can't be completely separated as the manuscript seems to suggest they can. Another problem is physical. There's a reason why statistical hydrology and climate studies of this sort normally focus on a common period of record, to ensure apples-to-apples comparisons. Especially in light of the fact that the minimum period of record seems to be a reasonable but not great 30 years, the observed trends could simply represent decadal (e.g. PDO) climate regime shifts, and because there's (apparently) different 30-year periods between the records, figuring out what these different trends at different stations actually means seems messy. There have been some studies that have gotten away with using (slightly) mixed periods of records, but these have been very geographically tightly focused studies of a dozen streamgages or less and evaluated those data in depth for very particular phenomena of interest. In such a large-scale pattern-recognition study as that presented here, though, it's not clear this works. A stronger case should be made for it, or at least a discussion giving better clarity on the pros and cons.

Lines 184-187, motivating the use of dynamic time warping: "The timing of inflections does not affect the clustering, hence the effects of latitude and elevation that often result in misclassification of hydrographs because of timing differences are avoided, which is important in a spatial domain of the size being considered here." That's fine as far as it goes, but the apparent down sides of such an approach should also be clarified for readers. Differences in timing and magnitude are basic identifying characteristics of a watershed's hydrologic regime and represent real differences in climate and watershed hydrology.

Lines 195-197, speaking of statistical trend analysis: "Since these were comparing periods separated by 360 days, autocorrelation was not expected and therefore pre-whitening was not applied." Careful here – there's a significant body of literature on the question of whether pre-whitening is or isn't needed in statistical analysis of streamflow

for climate trends. It seems like the verdict is still out, but some would view the authors' assertion are being a basic technical error, at least without further analysis to quantify autocorrelation across years. Again, it looks like some more detailed analysis and more careful wordsmithing are needed.

The explanation of dynamic time warping is inadequate (lines 122-133). This is a really interesting idea that has not seen much if any application in hydrology and is presented in the paper as one of its main elements. Much more detailed description of it would be useful for readers. Right now, there's just one sentence of explanation and a reference to the R package used to run it. It's also not clear how the cluster analysis the authors perform using dynamic time warping relates to the cluster analysis they perform using conventional k-means analysis. Are both used in the same way, as checks on each other? Or for slightly different purposes?

More broadly, the overall descriptions of the methods and their rationales feel a little murky, meandering, and internally inconsistent. The numerous typographical mistakes, grammatical errors, and poorly written sentences don't help, making the manuscript unnecessarily difficult to read and understand at points. It also feels like the paper is longer than it needs to be for its content, or maybe it is merely a matter of organization. Reorganizing the material more clearly into finer subsections with clearly worded and specific headings might help. The starting point would be to have separate "data" and "methods" sections. I suggest the authors rewrite the methodological explanation and justification from scratch and maybe consider a ground-up rewrite of the entire article focusing on making a clear case for the chief outcomes they want to communicate.

Lines 578-581, "Many studies that seek to explore change perform many analyses simply to see what falls out. The approach used here is targeted to assess which aspects are changing and why, the basic element being the hydrological response that depends upon the key streamflow-generating processes in each basin." Perhaps it was not the authors' intention, but this passage seems to come across like a somewhat ungracious and perhaps inaccurate description of earlier work in the hydrology and

climate communities. There are many change studies that have performed extremely well-designed and very tightly focused analyses. The authors could also be setting the bar rather high for themselves by framing their study relative to previous work in such elevated terms; it may be in their own best interest to reconsider that choice.

There are a couple of assertions in the paper, about a claimed resilience of mountain rivers to climate change, that seem likely to be controversial and may undermine its credibility as a whole with readers. Lines 812-817 say, "Using a modelling approach, (Bennett et al 2012; Schnorbus et al 2014) demonstrated that detecting climate driven changes in basins in the British Columbia Rocky Mountains were difficult because of interannual variability. Despite the ongoing deglaciation in the mountains of the west (Clarke et al 2015) basins in the Canadian Rockies can be resilient to change (Harder et al 2015, Whitfield and Pomeroy 2016)." Then again around line 923 or so we have, "Mountain basins appear to be resilient to change." These statements seem inconsistent with a lot of work in the region. A few examples are Jost et al, HESS, 2012; St. Jacques et al, Canadian Water Resour. J., 2014; St. Jacques et al, Geophys. Res. Letters, 2010; Fleming and Weber, J. Hydrol., 2013; Fleming and Dahlke, Canadian Water Resour. J., 2014; Najafi et al, Geophys. Res. Letters, 2017; and Clarke et al, Nature Geoscience, 2015. Additionally, I don't think Bennett et al 2012 and Schnorbus et al 2014 said quite what the authors of this submission seem to be ascribing to them, because both those papers clearly identified expected changes in hydrology from climate change. Plus, these assertions seem contrary to the general understanding in the hydrology and climate change communities that mountain regions are particularly susceptible to climate change, though of course such a general truism isn't necessarily applicable everywhere. At least part of the problem may be what seems to be a logical error in interpretation. On lines 923-927 we have, "These basins demonstrate several hydrograph types but generally lack structure in trend patterns. Individually, these basin do show periods with increases or decreases in streamflow consistent with freshet timing changes, as has been reported elsewhere, but there is sufficient inconsistency among the basins to define a specific pattern." I guess the idea here is that if there isn't

clear spatial consistency in trends, then there are no real trends, and the mountain region considered here is (therefore, in this view) resilient to change. But watershed properties influence hydrologic trends resulting from climate change, producing major trend variability between basins. Glaciers in alpine watersheds are a really well known example, and a relevant one in light of the fact that huge ice fields are at the headwaters of many rivers draining eastward off the Canadian Rocky Mountains and that are presumably considered in this article. For background and examples see Jansson et al, J. Hydrol., 2003; Fleming and Clarke, Can. Water Resour. J., 2003; Dahlke et al, HESS, 2012; Jost et al, HESS, 2012; Baraer et al, J. Glaciology, 2012; Fleming and Dahlke, Can. Water Resour. J., 2014; Moore et al, Hydrol. Proc., 2009; Stahl and Moore, Water Resour. Res., 2006; Stahl et al, Water Resour. Res., 2008; Fleming et al, Advances Water Resour., 2016; Casassa et al, Hydrol. Proc., 2009; Li et al, Hydrol. Proc., 2010; O'Neel et al, Climatic Change, 2014; O'Neel et al, Bioscience, 2015. Maybe the mountains of western Canada are, as the authors of this study suggest, more resilient to change than had previously been thought, but if so, it feels like a much more convincing case for it has to be made with complete and accurate referencing of the relevant literature.

The discussion of interannual/interdecadal climate variability like ENSO and the PDO on lines 829-837 is inadequate. The reference to these phenomena as "climate signals" is vague (climate change and other climate processes produce climate signals too), the passage is under-referenced, and the view it gives of these effects is not sufficiently incomplete.

It feels like the referencing around climatic changes that might be causing the hydrologic and landscape change inferred in this study might be improved a little. The most notable omissions include Vincent et al., J. Climate, 2015; and Vincent et al., Atmos.-Ocean, 2018.

"Section 4: Code Availability" is empty, and insufficient detail is provided for data sources in "Section 5: Data Availability."

[Figure]

While using sequent 5-day periods of the year for analysis makes sense, plotting them up in this way does not. Nobody intuitively thinks of seasons this way – what time of year is, say, the 23rd sequent 5-day interval? It makes interpretation of many of the figures and discussions in the manuscript unnecessarily opaque (Figure 8 is one example). A clearer case for using high-frequency data in long term change analysis could also be helpful. A starting point would be to provide more references to studies using variations of this approach like Hatcher and Jones, Atmos.-Ocean, 2013, Fleming et al, Advances Water Resour., 2016, and Vincent et al, Atmos.-Ocean, 2018.

The figures need work. Obvious examples are the confusing use of numbered sequent 5-day periods instead of day or month of year in many of the figures (see above comment), the lack of any axis labels on Figures 8 and 11, and the lack of even the most basic geographic details on Figure 6 for readers unfamiliar with Canada.

---

## Referee Comment (RC2) · Anonymous Referee #2 · 2 Mar 2020

General comments:

Paper The Spatial Extent of Hydrological and Landscape Changes across the Mountains and Prairies of the Saskatchewan and Mackenzie Basins examines spatial distribution of streamflow regime types, trend patterns and satellite indices (NDVI, NDWI, NDSI) based on large number of streamflow and satellite data sets covering large area of continental Canada, east of Continental divide. Main contributions of the paper are: (1) applications of methodology such as dynamic time-warping which enabled alignment of stream flow hydrographs according to the point of inflection and K-means clustering enabled classification of seasonal streamflow regimes; (2) large spatially

distributed data sets offering insights into changes in hydrological regimes and trend in large area covering several climate and topographical zones ; (3) increasing number of available datasets with applied methodology.

Overall this is very ambitious study done with the large data set covering large portion of continental Canada which offers new insights on hydrological changes in (especially) streamflow regimes and opens new research questions and deserves to be published in HESS. However, information and ideas presented in the paper are very difficult to follow so I recommend restructuring the text and adding some additional clarification to the questions presented in the next section. I recommend this article for final publication after the MAJOR revision, mainly regarding the paper structure and more concise communication of (very interesting and valuable) results.

Specific comments:

Remarks that should be addressed in order to make paper more concise are listed below. Three main open questions/remarks that need to be addressed are: 1) Different concepts are presented and used in the text: e.g. landscape and ecozones are used throughout text interchangeably. What is the difference between them? This comment is also related to the title – landscape is stressed in the title and in the paper, but analysis is done related to the ecozones maps. What was the main motivation for the introduction of ecozones and what additional information does it offer in the explanation of e.g. streamflow regime types and trend patterns? Although ecozones are connected with the climate and topography (and with analysed satellite indices), from the aspect of hydrological processes and streamflow regimes, watershed level is the most important unit that would offer additional insights (this is also stated by the authors in the paper Pg. 13 L 300-303, Pg. 21 L 580-581). Also, maps that give information about climate zones and topography of researched part of Canada would be more useful for analysis of results, especially about streamflow regime types and trend patterns, but also satellite indices. Authors should decide what would be the main goal and main information that they would like to convey in the paper and then should choose appropriate spatial

representations of the data. Question is again raised regarding landscape change and its influence on hydrological change on Pg. 30 L841 but answer or explanation is not given.

2) In the Pg. 6, L 140 it is stated that only time window between 19th April to 31st October for the streamflow data is used, and that satellite indices (NDVI, NDWI, NDSI) are extracted from the Landsat composite images for every sixteen days between 1980 and 2013 for the entire year (if I understood correctly). Although different time period is used for the hydrological storage (satellite) indices than for the streamflow regime and trend patterns, the reason why the same warm season time window of the data (between 19th April to 31st October) for the satellite indices is not used should be addressed. This would reduce the size of available data sets, but methodologically seasonal data would be comparable. Maybe this important methodological aspect of the paper, i.e. spatial analysis during the warm season, should be stressed and added to the title of the paper?

3) Methodology regarding dynamic time-warping and trend pattern analysis need additional clarification or at least more clear explanation of the idea and of the conducted steps.

Remaining questions / remarks:

4) After the introduction, I recommend adding one (sub)chapter named "Data" where more specific information would be given about used dataset and (sub)basins, before any processing of the data. After that (sub)chapter, chapter about used methods for processing and selection of the data could follow. Readers would especially benefit if the map with Saskatchewan and Mackenzie Basins location in Canada and location of analysed stations would be provided. Also, table with summary statistics of streamflow data collected from 395 basins would offer additionally information important for understanding of the analysed streamflow and watersheds (e.g. min, max, mean of analysed streamflow, dataset lengths, (area and mean elevation of analysed watersheds, etc.).

[Figure]

5) In Pg6, L144 - it is not clear for the reader what does "periods 23 to 61 (of 73)" represent. Also, this periods need to be marked appropriately in related figures (Fig 1-5,7,8, 11, S1-12, S14-20) and information about months or dates in the year (is it from 19th April to 31st October?) would be more useful. Especially since these are main figures for the understanding of the results presented in the paper.

6) Overall, information and ideas presented in the paper are difficult to follow so I recommend restructuring the text and careful reviewing of naming used: e.g. main spatial areas of change introduced in the discussion are [i] North of 60°; [ii] Boreal [iii] Prairies [iv] Mountains and in the conclusion are: [i] The Mackenzie Basin, [ii] The western Boreal Plains, [iii] the western Prairies, [iv] The Cordillera. Naming of the areas are different, and are also different from the three areas mentioned in the abstract: [i] north of 60°, [ii] in the western Boreal Plains, [iii] across the Prairie. Also, these areas should be mentioned and explained earlier in the discussion, not just to start section with this naming.

7) Abstract should be more concise, shortened and connected more with the main conclusions (one example just mentioned in the previous section). Questions opened in the abstract are very general (Pg. 5 L 103-108) and have not been answered in the discussion nor in the conclusion. Information about the data used in this study are presented in different sections of the text and it is difficult to follow what was available (e.g. in chapter 2.1. Data streamflow data and satellite data is introduced and 3 pages later in 2.3 Landscape and hydrological storage trends satellite data is introduced).

8) Pg. 6 L155-159 – it is not explained clearly enough why Figure 2 is important.

9) Pg. 7 L 175-176 "Only the data in the periods between the two vertical dashed lines in Figure 3 were used in the clustering". These lines are not marked consistently in the remaining Figures (both in text and supplemental S1-S12) and they are important for the understanding of the analysed period. 10) Results presented in Tables 5-7 show that fraction of stations showing a trend at $p \leq 0.05$ is decreasing with the number of

stations increasing. This would be interesting to comment in the text.

11) Introduction of the analysis of the recession limb of streamflow regime hydrographs is made for the first time in the discussion (Pg. 22 L601). This makes no sense because it has not been mentioned earlier as one of the goals of this research. Although this analysis offers new interesting insights, it should be introduced and explained earlier in the introduction and in the methodology.

12) It is not clear why would authors want to introduce questions regarding PDO and AO (Pg. 29 L829-836) and how is that connected and important with the results that they presented in their paper. What would be methodology used to incorporate these signals in their future work?

Technical corrections:

Figure 4 – 6 trend patterns (clusters) should be marked on vertical axis? Figure S2 - description of the figure should be cluster number 2, not 3? Figure S4 / S5 – it is not clear where does description of the Figure belong

---

## Author Comment (AC1) · 12 May 2020

Anonymous Referee #1 General comments: This is a really interesting, innovative, and valuable study that deserves to be published in HESS. The existing database of streamflow measurements in western Canada east of the continental divide is expanded, perhaps greatly, compared to earlier statistical studies of hydrologic change in the region by adding datasets from

streamgages that are only operated seasonally (i.e. not in winter). Some intriguing new analytical methods are introduced, in particular dynamic time warping. Further, Landsat imagery is interpreted alongside the statistical hydrology results, giving additional insights. The region itself also warrants close study in light of its size and, especially in its northern reaches, relative lack of study compared to many other places. Unfortunately, the manuscript doesn't seem to do a great job of communicating the work, and it has the feeling of being only half-finished, as explained in some detail in the specific comments below. I therefore recommended publication of this (potentially excellent) article pending major revisions that entail among other things an extensive ground-up rewrite.

Authors' response: We appreciate these comments and the reviewer's concerns. We have endeavoured to make substantive corrections and changes to address these concerns in the revised manuscript. The manuscript has been extensively rewritten, reorganized, figures have been modified and some additional figures and tables added to address this reviewer's comments and suggestions. —

Specific comments: The abstract feels clunky. There are unnecessary details (e.g. clarifying that the Landsat imagery analysis covers a different timeframe from the streamflow data analysis) while at the same time some of the big punch lines from the study seem to be missing.

Authors' response: We think that it is important to make the reader aware that the satellite imagery is for different periods of time than the streamflow. We have rewritten the abstract so that it is no longer 'clunky', at least in our opinion. As to the "Missing big punch lines", we chose to be careful to not overstate the results, and in revision have made several more forceful. —

There are odd writing choices (such as using both quotation marks and capitalization for "Streamflow Regime Types" and "Trend Patterns" when neither of these terms/concepts is new in any way). And some of it is just plain confusing.

[Figure]

Authors' response: We disagree with this comment. It is important that the reader understand when we are referring to members or either one of these two lists. In the revised abstract we have removed the quotation marks and capitalization but in the balance of the paper we use these to specifically refer to cluster/group membership. When the terms are introduced, quotes and capitals to highlight that we are using these terms in a very specific manner that is different from common practice. When used un-capitalized they are used with their common meaning. —

The use of data from rivers that are intermittent or ephemeral, because for example they're frozen completely or unmonitored in winter, is one of the more interesting and new aspects of this study. However, this dimension of the paper also feels like it hasn't been explored as much as it could and maybe should have. On the one hand, there may be a bit of missed opportunity here. The study doesn't seem to give much information on the main advantage of this approach, specifically, that it could hugely increase the sample size by pooling results across both perennial and intermittent/seasonal rivers; I don't see any numbers in the article about how much the available dataset is expanded by using this philosophy?

Authors' response: We agree that this is a key aspect of the paper since most studies only use stations with perennial flow and hence temporary streams are often overlooked. A classic trend study would have a few stations with perennial flow, and a common time window. The present study uses data from a large number of stations and includes both perennial and temporary streams, and does not use a common period of years. The revised manuscript provide details that address these points. Additional plots were added to the supplementary material to demonstrate missingness and dataset size for both continuous and seasonal stations. Text has been added to provide both more detail and more emphasis on the importance of this. —

And one of the most interesting questions that could be addressed in a study applying statistical change analysis to streamflow data from seasonal cold-regions rivers would be whether there have been changes in the annual no-flow duration and/or its dates of

onset, etc; that hasn't been tackled here at all as far as I can see.

Authors' response: While we agree that this would be interesting, investigating no flow duration and its onset is far more complicated than the reviewer may expect. Data suitable for a no flow duration associated with winter is rare as most seasonal stations cease operation at the end of October and do not start until the following April. No flow also occurs in the both the cold and warm season in many of the seasonal stations, but addressing it is beyond the scope of the present paper. —

At the same time, the approach used (if I understand it correctly) has the disadvantage of looking only at the common seasonal period between each streamgage time series, which means that wintertime flow measurements are not considered; that's the tradeoff.

Authors' response: This is the case. The common element used to compare data from these stations is a seasonal time window that is common to all stations, the "annual common time window". Where data exists outside the window as at some stations, analysis is reported at the station level but all comparisons use the same seasonal window. In several places we comment that members of a Trend Pattern with continuous data share the same trend pattern outside the common window. The use of a common seasonal time window not a disadvantage as it provide the opportunity to compare data from a much large number of locations. —

While some of the rivers in the study don't flow in winter, many do. Given that some of the stronger changes in climate and hydrology at high latitudes and cold regions generally are being experienced in winter, certain important phenomena could be missed using this approach. That fact doesn't necessarily reduce the "publishability" of the paper, but it does imply that a clearer statement of scope and purpose ought to be made. Maybe I've just missed it (a real possibility given the clunky writing and organization of the paper) but I don't see anywhere in the title or abstract that the study is limited to the warm season and it doesn't look like there's much discussion in the paper as a whole of what this limitation means.

[Figure]

Authors' response: In the revision we highlight the numbers of station and years included in this approach vs a common time period. We make the point in the abstract that the seasonal data is collected in the warm season. Could important phenomena be missed using this approach? This is probably the case for every trend study, but we clearly explain and limit what we are examining and how and others may wish to hypothesize further and conduct additional analysis. The increase in winter streamflow in the north is noted in the paper, and the fact that detecting trends in only the warm season created clusters where the winter streamflow was also increasing. Some text has been added to emphasize this point. —

Lines 100-103, phrases like "by integrating different forms of data, which previously had only been treated separately and individually" and "By linking continuous and partial year data from a large number of hydrometric stations using only warm season data three important questions are addressed. . ." I worry that many readers might feel that simply using data from summer, or looking at Landsat data alongside the streamflow data analysis, doesn't really count as "data integration," which might be taken to mean something more ambitious than what was done here. It might be safer to simply say that an unusually large and diverse dataset was considered.

Authors' response: We have made wording changes, where appropriate, to make the data availability clear. We more carefully describe the nature of the 'integration' that is achieved. We added a figure to the main text identifying the large basins and the names of important rivers and the locations of continuous and seasonally operated stations. Figures showing missingness and data availability, for continuous and seasonally operated stations were to the supplementary material so that the reader will be able to see the complete data picture; basically a common time window approach would be limited to a ~25 year period for only ~130 sites. —

Not clear what lines 134-135 are intended to mean; typographical error?

Authors' response: Indeed a typo. Should read: A threshold of p<0.05 is used in tests

of significance, and 5% is used as an indicator that the number of trends exceeds the number that would be expected by chance alone. —

The idea of not using a common period of record across all the streamgages runs contrary to almost all work in this field. That doesn't necessarily mean it's wrong, but it does seems likely to be viewed as a problem by some readers, and the subject probably requires more attention than it's been given in the article. For example, the authors write on lines 138-144 that "Because the periods of record, rather than a common period, are used it is not possible to compare the magnitudes of trends among the stations. Instead, the analyses are restricted to determining the existence of significant trends in individual five-day periods in (five-day) periods 23 to 61 (of 73), as shown in Figure 1 and 2." In addition to being a clunky and hard-to-read sentence, its logic seems doubtful. If the magnitudes of trends are not comparable between stations, why would the existences of trends be? One problem is statistical. Determining the existence of trend using a statistical significance test (as done here) simply amounts to a measure of the magnitude of the observed trend relative to what you'd expect to see purely by chance; the concepts of magnitude and existence can't be completely separated as the manuscript seems to suggest they can. Another problem is physical. There's a reason why statistical hydrology and climate studies of this sort normally focus on a common period of record, to ensure apples-to-apples comparisons. Especially in light of the fact that the minimum period of record seems to be a reasonable but not great 30 years, the observed trends could simply represent decadal (e.g. PDO) climate regime shifts, and because there's (apparently) different 30-year periods between the records, figuring out what these different trends at different stations actually means seems messy. There have been some studies that have gotten away with using (slightly) mixed periods of records, but these have been very geographically tightly focused studies of a dozen streamgages or less and evaluated those data in depth for very particular phenomena of interest. In such a large-scale pattern-recognition study as that presented here, though, it's not clear this works. A stronger case should be made for it, or at least a discussion giving better clarity on the pros and cons.

Authors' response: The main response to this comment is that this is indeed the novelty of this manuscript. We totally agree that if the interest was in determining trend magnitudes that a common time window and complete annual records would be used. Even if this is the case, however, the trends that are detected may be inappropriately interpreted as trends over time result from complicated changes within the climate system and include shifts in climate indices [AMO, NAO, PDO, and ENSO] that confound attribution. In the present study we simply focus on whether or not the five day periods on the common time window demonstrate a coherent trend pattern. Stations may not show a pattern for many reasons, the available record may be too short in some cases, but the trends reported are those present in each dataset. The reviewer should understand that this approach is also an apples to apples comparison just different to how many studies are done. We accept that our approach may be perceived as contrary to much of the work in the past. While we agree that it would be nice to have the most common apples to apples comparison even a common time window does no guarantee that that is achieved. Selecting a particular block of years does not guarantee that any trends that are detected are attributable to warming climate since the climate system is complex and variable. In our approach we focus on changes in the seasonal pattern of streamflow and whether those patterns have a spatial pattern. We disagree with the reviewer with respect to their point on separating trend magnitude and direction. These are not completely coupled. It is reasonable to separate direction from magnitude; in fact asking the question "Is there a trend?" does not imply a need to know how large. The nonparametric Mann-Kendall is used to analyze whether a series is consistently [monotonically] increasing or decreasing but itself does not resolve magnitude. To be able to compare magnitudes it would indeed need to have a different apples to apples basis. This has been noted in the text. —

Lines 184-187, motivating the use of dynamic time warping: "The timing of inflections does not affect the clustering, hence the effects of latitude and elevation that often result in misclassification of hydrographs because of timing differences are avoided, which is important in a spatial domain of the size being considered here." That's fine

as far as it goes, but the apparent down sides of such an approach should also be clarified for readers. Differences in timing and magnitude are basic identifying characteristics of a watershed's hydrologic regime and represent real differences in climate and watershed hydrology.

Authors' response: The text has been expanded to and clarified. The point is that a common process, for example snowmelt, has different timing with latitude and elevation, and therefore classifications need to not confuse generating processes because of timing differences. —

Lines 195-197, speaking of statistical trend analysis: "Since these were comparing periods separated by 360 days, autocorrelation was not expected and therefore prewhitening was not applied." Careful here – there's a significant body of literature on the question of whether pre-whitening is or isn't needed in statistical analysis of streamflow for climate trends. It seems like the verdict is still out, but some would view the

Authors' assertion are being a basic technical error, at least without further analysis to quantify autocorrelation across years. Again, it looks like some more detailed analysis and more careful wordsmithing are needed.

Authors' response: There is indeed a large body of literature on prewhitening. It is very clear that when data are closely spaced in time such as daily and hourly observations, or when the time series is assembled from large numbers of samples [annual mean streamflow] the data may be expected to be autocorrelated. It is much less clear of how to apply it in datasets where autocorrelation is low and many cases not significant. We have examined this for many cases over the past decades; in most rivers only about 15% of the cases show autocorrelation when more than 100 years of data are available. With 60 years of data less than 10% of the cases have significant autocorrelation. Just for clarity, we took rivers with long records and sampled them every five days and calculated the autocorrelation for each of the 73 'slices' and compared those to the

level of significance determining the proportion that exceed the significance threshold (∼0.19). The autocorrelations that do exceed the thresholds are small (∼0.2 to 0.25). So, in our opinion it is better to treat all the datasets in the same fashion and not introduce additional complications either by prewhitening all the cases because a few cases have signification autocorrelation or prewhitening some and not the majority. A few lines have been added to the manuscript to reflect this. —

The explanation of dynamic time warping is inadequate (lines 122-133). This is a really interesting idea that has not seen much if any application in hydrology and is presented in the paper as one of its main elements. Much more detailed description of it would be useful for readers. Right now, there's just one sentence of explanation and a reference to the R package used to run it. It's also not clear how the cluster analysis the authors perform using dynamic time warping relates to the cluster analysis they perform using conventional k-means analysis. Are both used in the same way, as checks on each other? Or for slightly different purposes?

Authors' response: We have added some additional text and an additional component to the Figure to better explain the method. —

More broadly, the overall descriptions of the methods and their rationales feel a little murky, meandering, and internally inconsistent. The numerous typographical mistakes, grammatical errors, and poorly written sentences don't help, making the manuscript unnecessarily difficult to read and understand at points. It also feels like the paper is longer than it needs to be for its content, or maybe it is merely a matter of organization. Reorganizing the material more clearly into finer subsections with clearly worded and specific headings might help. The starting point would be to have separate "data" and "methods" sections. I suggest the authors rewrite the methodological explanation and justification from scratch and maybe consider a ground-up rewrite of the entire article focusing on making a clear case for the chief outcomes they want to communicate.

Authors' response: We revised the methods to provide the reader with more information

about the data and analysis. —

Lines 578-581, "Many studies that seek to explore change perform many analyses simply to see what falls out. The approach used here is targeted to assess which aspects are changing and why, the basic element being the hydrological response that depends upon the key streamflow-generating processes in each basin." Perhaps it was not the Authors' intention, but this passage seems to come across like a somewhat ungracious and perhaps inaccurate description of earlier work in the hydrology and climate communities. There are many change studies that have performed extremely well-designed and very tightly focused analyses. The authors could also be setting the bar rather high for themselves by framing their study relative to previous work in such elevated terms; it may be in their own best interest to reconsider that choice.

Authors' response: While it might be ungracious, this is a classic problem in the literature despite there being many well designed studies. We have rewritten the text to make the point in a more gracious manner. —

There are a couple of assertions in the paper, about a claimed resilience of mountain rivers to climate change, that seem likely to be controversial and may undermine its credibility as a whole with readers. Lines 812-817 say, "Using a modelling approach, (Bennett et al 2012; Schnorbus et al 2014) demonstrated that detecting climate driven changes in basins in the British Columbia Rocky Mountains were difficult because of interannual variability. Despite the ongoing deglaciation in the mountains of the west (Clarke et al 2015) basins in the Canadian Rockies can be resilient to change (Harder et al 2015, Whitfield and Pomeroy 2016)." Then again around line 923 or so we have, "Mountain basins appear to be resilient to change." These statements seem inconsistent with a lot of work in the region. A few examples are Jost et al, HESS, 2012; St. Jacques et al, Canadian Water Resour. J., 2014; St. Jacques et al, Geophys. Res. Letters, 2010; Fleming and Weber, J. Hydrol., 2013; Fleming and Dahlke, Canadian Water Resour. J., 2014; Najafi et al, Geophys. Res. Letters, 2017; and Clarke et al, Nature Geoscience, 2015. Additionally, I don't think Bennett et al

2012 and Schnorbus et al 2014 said quite what the authors of this submission seem to be ascribing to them, because both those papers clearly identified expected changes in hydrology from climate change.

Authors' response: We modified wordings to specifically relate this to our results. While we disagree with the reviewer's comment about our interpretation of the papers by Bennett et al and Schnorbus et al. we did change 'interannual variability' to "uncertainty". Bennett et al (2012) state that based on their model projections "For the 2050s time period, runoff anomalies examined for winter had the largest range in GCMs over other seasons, emissions scenarios, and the hydrologic parameterizations, in that order, with an average range for all basins of 84%, 58%, and 31%, respectively." Schnorbus et al 2014 state in their conclusions that "There is a high degree of consensus for a decrease in summer precipitation in southern BC (CR and UCR). Summer precipitation results for north-eastern BC (the PR) are more ambiguous." Versus. "Runoff changes in the three areas vary spatially and seasonally and generally reflect snow and precipitation changes and topography based temperature gradients." And "The degree to which the MD site transitions from a glacial–nival regime to strictly nival regime by the mid-21st century is potentially underestimated in the current study." Additional text has been provided to specifically indicate that the changes we observe in the mountains are different from other relevant published work. —

Plus, these assertions seem contrary to the general understanding in the hydrology and climate change communities that mountain regions are particularly susceptible to climate change, though of course such a general truism isn't necessarily applicable everywhere. At least part of the problem may be what seems to be a logical error in interpretation. On lines 923-927 we have, "These basins demonstrate several hydrograph types but generally lack structure in trend patterns. Individually, these basin do show periods with increases or decreases in streamflow consistent with freshet timing changes, as has been reported elsewhere, but there is sufficient inconsistency among the basins to define a specific pattern." I guess the idea here is that if there isn't clear
spatial consistency in trends, then there are no real trends, and the mountain region considered here is (therefore, in this view) resilient to change. But watershed properties influence hydrologic trends resulting from climate change, producing major trend variability between basins. Glaciers in alpine watersheds are a really well known example, and a relevant one in light of the fact that huge ice fields are at the headwaters of many rivers draining eastward off the Canadian Rocky Mountains and that are presumably considered in this article. For background and examples see Jansson et al, J. Hydrol., 2003; Fleming and Clarke, Can. Water Resour. J., 2003; Dahlke et al, HESS, 2012; Jost et al, HESS, 2012; Baraer et al, J. Glaciology, 2012; Fleming and Dahlke, Can. Water Resour. J., 2014; Moore et al, Hydrol. Proc., 2009; Stahl and Moore, Water Resour. Res., 2006; Stahl et al, Water Resour. Res., 2008; Fleming et al, Advances Water Resour., 2016; Casassa et al, Hydrol. Proc., 2009; Li et al, Hydrol. Proc., 2010; O'Neel et al, Climatic Change, 2014; O'Neel et al, Bioscience, 2015. Maybe the mountains of western Canada are, as the authors of this study suggest, more resilient to change than had previously been thought, but if so, it feels like a much more convincing case for it has to be made with complete and accurate referencing of the relevant literature.

Authors' response: In our opinion, there is no logical error. Warming in mountainous areas has the potential to result in large hydrologic change and that has been widely reported as the reviewer indicates. We make the case that when comparing stations that are closely located in space, with either similar or dissimilar streamflow generating mechanisms, one should expect that the seasonal pattern of changes would be similar. This is a different perspective to many of the papers the reviewer lists. There is no question that glaciers are melting, but this does not always result in changes in seasonal patterns of streamflow. The fact that there is inconsistency is what is important; however, that does not mean that there are no trends. We added some text to emphasize these differences. —

The discussion of interannual/interdecadal climate variability like ENSO and the PDO

on lines 829-837 is inadequate. The reference to these phenomena as "climate signals" is vague (climate change and other climate processes produce climate signals too), the passage is under-referenced, and the view it gives of these effects is not sufficiently incomplete.

Authors' response: We disagree. We acknowledge in the discussion that these signals were not considered in this study but that some authors had reported effects of PDO and AO.

It feels like the referencing around climatic changes that might be causing the hydrologic and landscape change inferred in this study might be improved a little. The most notable omissions include Vincent et al., J. Climate, 2015; and Vincent et al., Atmos.-Ocean, 2018.

Authors' response: Vincent et al (2015) studied the trends in temperature and precipitation associated with low frequency modes which we acknowledged we did not investigate. We have included text to reflect the changes in snowfall days reported in Vincent et al 2018. —

"Section 4: Code Availability" is empty, and insufficient detail is provided for data sources in "Section 5: Data Availability."

Authors' response: These headers were required by HESS even though they are not used. The access to the data through ECDataExplorer is described in the main text. This should be sufficient for others to access the data in the same manner. —

While using sequent 5-day periods of the year for analysis makes sense, plotting them up in this way does not. Nobody intuitively thinks of seasons this way – what time of year is, say, the 23rd sequent 5-day interval? It makes interpretation of many of the figures and discussions in the manuscript unnecessarily opaque (Figure 8 is one example).

Authors' response: We think that the reviewer is simply asking for information about

the months of the year to compliment the series of 5-day periods. We have added secondary axes to all the plots in the main paper that show the months of the year, and a table to show the starting dates of each 5-day period, but the series of 5 day periods upon which the analysis is based are retained and used as the point of reference. —

A clearer case for using high-frequency data in long term change analysis could also be helpful. A starting point would be to provide more references to studies using variations of this approach like Hatcher and Jones, Atmos.-Ocean, 2013, Fleming et al, Advances Water Resour., 2016, and Vincent et al, Atmos.-Ocean, 2018.

Authors' response: We do not understand what the reviewer intends by "A clearer case for using high-frequency data in long term change analysis could also be helpful." References to Hatcher and Jones (2013) regarding resilience in Columbia headwaters and Vincent et al (2018) regarding changes in days with snow were added into the text. Fleming et al (2016) deals with teleconnections which, as noted above, we acknowledge we did not address. —

The figures need work. Obvious examples are the confusing use of numbered sequent 5-day periods instead of day or month of year in many of the figures (see above comment), the lack of any axis labels on Figures 8 and 11, and the lack of even the most basic geographic details on Figure 6 for readers unfamiliar with Canada.

Authors' response: A secondary axis indicating the months of the year was added to figures in the revised main and supplementary texts. A summary of all the changes to figures is provided at the end of this document. —

---

## Author Comment (AC2) · 12 May 2020

General comments: Paper The Spatial Extent of Hydrological and Landscape Changes across the Mountains and Prairies of the Saskatchewan and Mackenzie Basins examines spatial distribution of streamflow regime types, trend patterns and satellite indices

(NDVI, NDWI, NDSI) based on large number of streamflow and satellite data sets covering large area of continental Canada, east of Continental divide. Main contributions of the paper are: (1) applications of methodology such as dynamic time-warping which enabled alignment of stream flow hydrographs according to the point of inflection and K-means clustering enabled classification of seasonal streamflow regimes; (2) large spatially distributed data sets offering insights into changes in hydrological regimes and trend in large area covering several climate and topographical zones ; (3) increasing number of available datasets with applied methodology. Overall this is very ambitious study done with the large data set covering large portion of continental Canada which offers new insights on hydrological changes in (especially) streamflow regimes and opens new research questions and deserves to be published in HESS. However, information and ideas presented in the paper are very difficult to follow so I recommend restructuring the text and adding some additional clarification to the questions presented in the next section. I recommend this article for final publication after the MAJOR revision, mainly regarding the paper structure and more concise communication of (very interesting and valuable) results.

Specific comments: Remarks that should be addressed in order to make paper more concise are listed below. Three main open questions/remarks that need to be addressed are: 1) Different concepts are presented and used in the text: e.g. landscape and ecozones are used throughout text interchangeably. What is the difference between them? This comment is also related to the title – landscape is stressed in the title and in the paper, but analysis is done related to the ecozones maps. What was the main motivation for the introduction of ecozones and what additional information does it offer in the explanation of e.g. streamflow regime types and trend patterns? Although ecozones are connected with the climate and topography (and with analysed satellite indices), from the aspect of hydrological processes and streamflow regimes, watershed level is the most important unit that would offer additional insights (this is also stated by the authors in the paper Pg. 13 L 300-303, Pg. 21 L 580-581). Also, maps that give information about climate zones and topography of researched part

of Canada would be more useful for analysis of results, especially about streamflow regime types and trend patterns, but also satellite indices. Authors should decide what would be the main goal and main information that they would like to convey in the paper and then should choose appropriate spatial representations of the data. Question is again raised regarding landscape change and its influence on hydrological change on Pg. 30 L841 but answer or explanation is not given.

Authors' response: We chose to use ecozones as opposed to climate zones as reflective of the Canadian landscape. Climate zones are indeed intimately linked into the ecozones. We believe that ecozones are the appropriate context to examine these changes at this scale as that is where the changes are taking place. For example, Whitfield et al (2020) examined ecoregions and hydrological change in the Canadian Prairies; however, ecoregions would be too fine a scale to consider for ∼400 watersheds. We have included topography in a new figure that shows the locations of seasonal and continuous station, river basin boundaries, and names of major rivers. In revising the manuscript, we have paid specific attention to making linkages between hydrological change and changes in basins. —

2) In the Pg. 6, L 140 it is stated that only time window between 19th April to 31st October for the streamflow data is used, and that satellite indices (NDVI, NDWI, NDSI) are extracted from the Landsat composite images for every sixteen days between 1980 and 2013 for the entire year (if I understood correctly). Although different time period is used for the hydrological storage (satellite) indices than for the streamflow regime and trend patterns, the reason why the same warm season time window of the data (between 19th April to 31st October) for the satellite indices is not used should be addressed. This would reduce the size of available data sets, but methodologically seasonal data would be comparable. Maybe this important methodological aspect of the paper, i.e. spatial analysis during the warm season, should be stressed and added to the title of the paper?

Authors' response: We have added a comment regarding the choice of a different

seasonal window for satellite imagery than hydrological data. Satellite imagery is a different data type with images only available every eight days at best and adjusting to the time window of the hydrological data. The title has been modified to include 'warm season time window' —

3) Methodology regarding dynamic time-warping and trend pattern analysis need additional clarification or at least more clear explanation of the idea and of the conducted steps. Remaining questions / remarks:

Authors' response: Additional text and an additional panel in Figure 4 were added to provide the clarification. —

4) After the introduction, I recommend adding one (sub)chapter named "Data" where more specific information would be given about used dataset and (sub)basins, before any processing of the data. After that (sub)chapter, chapter about used methods for processing and selection of the data could follow. Readers would especially benefit if the map with Saskatchewan and Mackenzie Basins location in Canada and location of analysed stations would be provided. Also, table with summary statistics of streamflow data collected from 395 basins would offer additionally information important for understanding of the analysed streamflow and watersheds (e.g. min, max, mean of analysed streamflow, dataset lengths, (area and mean elevation of analysed watersheds, etc.).

Authors' response: We have separated the material describing the data from the analysis within the Methods section. An additional Figure was added to the main text showing the location of stations, the major river basins and the names of major rivers. An additional figure was added to the supplementary material showing the missingness/availability of data from continuous and seasonal sites and the overall annual data density. These address the comment about data lengths. Table 1 has been expanded to provide more details about the numbers of stations in the different major river basins. Some text was added to indicate the range of basin areas and station elevations. It is our opinion that the min, max, and mean of streamflow could be easy

to misinterpret and as we use z-scores the sites can be compared based upon those, since this provides a mean of zero and a range in standard deviations. A note to this effect has been inserted in the main text. —

5) In Pg6, L144 - it is not clear for the reader what does "periods 23 to 61 (of 73)" represent. Also, this periods need to be marked appropriately in related figures (Fig 1-5,7,8, 11, S1-12, S14-20) and information about months or dates in the year (is it from 19th April to 31st October?) would be more useful. Especially since these are main figures for the understanding of the results presented in the paper.

Authors' response: All figures showing the five-day periods now have a secondary month axis included. Thank you for noticing that the lines showing periods differed between some simple plots and raster figures. These are now consistent. —

6) Overall, information and ideas presented in the paper are difficult to follow so I recommend restructuring the text and careful reviewing of naming used: e.g. main spatial areas of change introduced in the discussion are [i] North of 60âŮȩ ; [ii] Boreal [iii] Prairies [iv] Mountains and in the conclusion are: [i] The Mackenzie Basin, [ii] The western Boreal Plains, [iii] the western Prairies, [iv] The Cordillera. Naming of the areas are different, and are also different from the three areas mentioned in the abstract: [i] north of 60âŮȩ , [ii] in the western Boreal Plains, [iii] across the Prairie. Also, these areas should be mentioned and explained earlier in the discussion, not just to start section with this naming.

Authors' response: We have addressed this by changing the text to use the same names throughout the manuscript. We now use North of $60°$, The Boreal Plains, the Prairies, and the Mountains in the same sense throughout the manuscript. —

7) Abstract should be more concise, shortened and connected more with the main conclusions (one example just mentioned in the previous section). Questions opened in the abstract are very general (Pg. 5 L 103-108) and have not been answered in the discussion nor in the conclusion. Information about the data used in this study are

presented in different sections of the text and it is difficult to follow what was available (e.g. in chapter 2.1. Data streamflow data and satellite data is introduced and 3 pages later in 2.3 Landscape and hydrological storage trends satellite data is introduced).

Authors' response: The description of the satellite data was moved to the section on data. —

8) Pg. 6 L155-159 – it is not explained clearly enough why Figure 2 is important.

Authors' response: This is a useful suggestion and additional text has been incorporated to explain this. —

9) Pg. 7 L 175-176 "Only the data in the periods between the two vertical dashed lines in Figure 3 were used in the clustering". These lines are not marked consistently in the remaining Figures (both in text and supplemental S1-S12) and they are important for the understanding of the analysed period.

Authors' response: Corrected as suggested. —

10) Results presented in Tables 5-7 show that fraction of stations showing a trend at $p \leq 0.05$ is decreasing with the number of stations increasing. This would be interesting to comment in the text.

Authors' response: A comment has been added to better explain this. —

11) Introduction of the analysis of the recession limb of streamflow regime hydrographs is made for the first time in the discussion (Pg. 22 L601). This makes no sense because it has not been mentioned earlier as one of the goals of this research. Although this analysis offers new interesting insights, it should be introduced and explained earlier in the introduction and in the methodology.

Authors' response: This has been addressed by introducing it in the methods. Thanks for pointing this out. But, the goals of the research did state "how are the hydrological types and processes distributed" and this material addresses that goal. —

12) It is not clear why would authors want to introduce questions regarding PDO and AO (Pg. 29 L829-836) and how is that connected and important with the results that they presented in their paper. What would be methodology used to incorporate these signals in their future work?

Authors' response: We felt that it was important to acknowledge that we did not consider the effects of PDO, ENSO, or AMO which others have identified to affect streamflow in the section of the discussion dealing with limitations. We have considered how to express this differently and incorporate it into the paper. [Another option was to simply remove any mention of them and be criticized for not covering them.] —

Technical corrections: Figure 4 – 6 trend patterns (clusters) should be marked on vertical axis?

Authors' response: Figure 4 shows an example of dynamic time warping where the y axis is labelled as discharge as z-score. We think the reviewer may mean Figure 8 to which we will add numbers to indicate the groups. —

Figure S2 - description of the figure should be cluster number 2, not 3?

Authors' response: This typographical error has been fixed. —

Figure S4 / S5 – it is not clear where does description of the Figure belong

Authors' response: A "new page" was inserted before Figure S5 so all four occur on the same page. —

Authors comments:

Other changes made in revision

Tables:

[Figure]

Table 1 was extensively revised to provide more detail about the distribution of stations between major watersheds.

Figures:

New Figure 1 was added to indicate locations of seasonal and continuous stations, the major river basin boundaries, the names of major rivers and the topography.

Figure 1 – a second axis showing months has been added.

Figure 2 – a second axis showing months has been added.

Figure 3 – a second axis showing months has been added.

Figure 4 - has been modified to include a figure showing the dtw alignment step.

Figure 7 – a second axis showing months has been added.

Figure 8 – a second axis showing months has been added, the separation between clusters has been made clearer and the cluster (Trend Pattern) shown in italics.

Figure 11 – labels added to axes

Supplementary Figures:

Captions of all supplementary figures were edited for clarity and consistency.

A New Figure S1 was inserted. The four panels show the missingness and the annual density of available station data for continuous and seasonal stations.

Figures S1-S12 – a second axis showing months has been added to each panel.

Figure S14 - a second axis showing months has been added.

Figures S15 to S20– a second axis showing months has been added to the raster portion and the layout of the figure elements were changed.
* * *
671, 2020.